# EdgeMask-DG*: Learning Domain-Invariant Graph Structures via Adversarial Edge Masking

**Rishabh Bhattacharya**                                          *rishabh.bhattacharya@research.iiit.ac.in*
*Machine Learning Lab @ IIIT-H*
*Hyderabad, India*

**Naresh Manwani**                                                      *naresh.manwani@iiit.ac.in*
*Machine Learning Lab @ IIIT-H*
*Hyderabad, India*

**Reviewed on OpenReview:** *https://openreview.net/forum?id=vkfe8Ke7eC*

## Abstract

Structural shifts pose a significant challenge for graph neural networks, as graph topology acts as a covariate that can vary across domains. Existing domain generalization methods rely on fixed structural augmentations or training on globally perturbed graphs, mechanisms that do not pinpoint which specific edges encode domain-invariant information. We argue that domain-invariant structural information is not rigidly tied to a single topology but resides in the consensus across multiple graph structures derived from topology and feature similarity. To capture this, we first propose EdgeMask-DG, a novel min-max algorithm where an edge masker learns to find worst-case continuous masks subject to a sparsity constraint, compelling a task GNN to perform effectively under these adversarial structural perturbations. Building upon this, we introduce EdgeMask-DG*, an extension that applies this adversarial masking principle to an enriched graph. This enriched graph combines the original topology with feature-derived edges, allowing the model to discover invariances even when the original topology is noisy or domain-specific. EdgeMask-DG* is the first to systematically combine adaptive adversarial topology search with feature-enriched graphs. We provide a formal justification for our approach from a robust optimization perspective. We demonstrate that EdgeMask-DG* achieves strong overall performance on diverse graph domain generalization benchmarks, including citation networks, social networks, and temporal graphs, with the best average results on the citation benchmark and competitive performance on several additional settings. Notably, on the Cora OOD benchmark, EdgeMask-DG* lifts the worst-case domain accuracy to 78.0%, a +3.8 pp improvement over the prior state of the art (74.2%). The source code for our experiments can be found here: `https://github.com/rbSparky/TMLR`

## 1 Introduction

Graph Neural Networks (GNNs) (Kipf & Welling, 2017; Veličković et al., 2018; Hamilton et al., 2017; Xu et al., 2019) have revolutionized learning on graph data, driving progress in fields ranging from social network analysis and bioinformatics to recommendation systems (Wu et al., 2021). However, their remarkable performance often relies on the stringent assumption that training and testing data are drawn from the same distribution (I.I.D.). In many real-world applications, this assumption is violated; data often originates from heterogeneous sources or evolves, leading to distribution shifts that can severely degrade GNN performance (Shi et al., 2024; Zhu et al., 2021a). Graph Domain Generalization (Graph-DG) aims to tackle this fundamental problem by developing models capable of generalizing from one or more source graph domains to unseen target domains with different underlying distributions.

A particularly challenging yet common scenario in Graph-DG involves structural distribution shifts (Chen et al., 2025; Wu et al., 2022). In tasks like cross-graph node classification (e.g., predicting research areas

of papers across different citation networks), the node features and label semantics often remain consistent, while the graph topology can vary drastically. GNNs trained naively via Empirical Risk Minimization (ERM) on source graphs overfit to these domain-specific topological patterns, failing to generalize to target graphs with different structures.

Existing approaches for Graph-DG include designing invariant architectures (Li et al., 2022), employing robust training objectives (Wu et al., 2022), or utilizing data augmentation (Chen et al., 2025). Data augmentation methods targeting structural shifts, such as GraphAug (Chen et al., 2025), often apply fixed heuristic rules (e.g., edge dropping based on node degrees, edge addition based on feature similarity). These static strategies may not optimally identify the domain-invariant structural factors, which can be complex and data-dependent.

To address these limitations, we first introduce the core algorithmic concept of **EdgeMask-DG**, a novel min-max adversarial framework. In EdgeMask-DG, an "edge masker" network learns to generate sparse binary or continuous masks over the edges of a given graph. This masker is trained adversarially against a "task GNN" to find masks that maximally disrupt the task GNN's performance, subject to a sparsity constraint on the mask. The task GNN, in turn, must learn to perform its task robustly under these worst-case structural perturbations. This process encourages the task GNN to rely on structural patterns that are inherently resilient to such adversarial masking.

However, relying solely on the original graph topology, even with adaptive masking, can be problematic if the source topology is sparse, noisy, or highly domain-specific. To overcome this, we propose an extension, **EdgeMask-DG\***, which applies the EdgeMask-DG adversarial masking principle to an enriched graph representation. This enriched graph is constructed by augmenting the original edges with new edges derived from node feature similarity, specifically using k-Nearest Neighbours (kNN) and spectral clustering. This allows EdgeMask-DG\* to leverage the (often assumed) invariance of the feature distribution ($P(\mathbf{X})$) to discover robust structures that might not be apparent in the original topology alone. The adversarial game then proceeds on this enriched graph: the MaskNet identifies challenging sparse masks over both original and feature-derived edges, and the TaskNet (implemented using a Graph Attention Network, GAT (Veličković et al., 2018)) learns to perform under these conditions.

A GAT backbone is well-suited for this task, as its layers can directly incorporate the learned edge masks as edge attributes, directly influencing message passing and attention. This dynamic, learned masking on an enriched graph allows EdgeMask-DG\* to adaptively identify and exploit the most reliable structural information for generalization.

Our contributions are:

- We propose EdgeMask-DG, a novel min-max adversarial learning framework for Graph-DG that learns domain-invariant edge masks adaptively on a given graph structure.

- We introduce EdgeMask-DG\*, an extension that applies this adversarial masking principle to an enriched graph structure, integrating original topology with feature-derived edges (kNN and spectral clustering) to leverage feature invariance and overcome limitations of the original topology.

- We employ a GAT backbone that naturally incorporates the learned continuous edge masks as attributes, enhancing the model's ability to focus on relevant substructures.

- We demonstrate through extensive experiments that EdgeMask-DG\* attains the best average performance on the citation Graph-DG benchmark and strong, competitive results across additional datasets spanning citation, social, and temporal graphs.

## 2   Related Work

**Graph Domain Generalization (Graph-DG).** Generalizing GNNs to out-of-distribution data is a critical challenge (Shi et al., 2024; Zhu et al., 2021a). Early work adapted principles like invariant risk minimization (Arjovsky et al., 2019) and distributionally robust optimization (e.g., EERM (Wu et al., 2022)). Other strategies include disentangling representations (Zhu et al., 2021b; Yu et al., 2023), filtering spurious

correlations via information-theoretic objectives (Yang et al., 2023), and unsupervised contrastive learning (Zhu et al., 2024). On the data augmentation front, methods range from static perturbations like GraphAug (Chen et al., 2025) to more advanced techniques like generating continuous invariant subgraphs (GRM (Wang et al., 2025)) or combining adversarial learning with mixup (TRACI (Zhao et al., 2025)). Our method differs by using adversarial edge masking to adaptively learn domain-invariant substructures rather than relying on predefined heuristics or variational objectives.

**Graph Data Augmentation (GDA).** GDA techniques aim to improve GNN robustness and generalization, often in standard graph learning settings (You et al., 2020; Zhu et al., 2021b). Common strategies involve node dropping, edge perturbation (random or heuristic addition/deletion), feature masking, and subgraph sampling (You et al., 2020). GraphAug (Chen et al., 2025) specifically targets Graph-DG structural shifts with predefined rules for edge dropping (low-weight) and edge adding (spectral clustering); however, its heuristics are static. Other works explore learnable GDA (Zhao et al., 2021), but typically optimize for source domain performance, which may not guarantee OOD generalization. EdgeMask-DG* uses an adversarial objective to learn domain-invariant structural masks over an enriched graph space.

**Adversarial Learning on Graphs.** Adversarial techniques are prevalent in graph learning, primarily for enhancing robustness against adversarial attacks designed to fool GNNs (Zügner et al., 2018; Dai et al., 2018). Adversarial domain adaptation (ADA) methods (Wu et al., 2020; Ma et al., 2019) use domain discriminators to align representations across domains, but typically require access to target domain data (labeled or unlabeled) during training, violating the DG setup. EdgeMask-DG* employs a distinct adversarial mechanism: the adversary (MaskNet) operates solely on source domains, perturbing the structure via learned masks to enforce generalizable robustness in the primary model (TaskNet), rather than defending against attacks or adapting to a specific target.

**Graph Attention Networks (GAT).** GAT (Veličković et al., 2018) introduced an attention mechanism allowing nodes to weigh the importance of their neighbours during message passing. Its ability to handle weighted graphs or incorporate edge features makes it well-suited for EdgeMask-DG*, where the learned mask values **s** act as dynamic edge attributes guiding the aggregation process. This contrasts with GCN (Kipf & Welling, 2017) or GIN (Xu et al., 2019), which typically treat edges uniformly or require modifications to handle edge weights effectively.

## 3   Preliminaries

**Notation.** We represent a graph as $\mathcal{G} = (\mathcal{V}, \mathcal{E}, \mathbf{X})$, where $\mathcal{V}$ is the set of $N = |\mathcal{V}|$ nodes, $\mathcal{E}$ is the set of edges, and $\mathbf{X} \in \mathbb{R}^{N \times d}$ is the node feature matrix with dimension $d$. The graph topology can be represented by an adjacency matrix $\mathbf{A} \in \{0, 1\}^{N \times N}$. Node labels are denoted by $\mathbf{Y}$. A GNN model is denoted by $f(\cdot)$, parameterized by $\theta$. The classification loss (e.g., cross-entropy) is $\mathcal{L}_{cls}$.

**Graph Domain Generalization (Graph-DG).** We are given $M$ source domains $\{\mathcal{G}_S^i\}_{i=1}^M$, each associated with a distribution $P_S^i(\mathbf{X}, \mathbf{Y}, \mathbf{A})$. The goal is to learn a model $f$ using data from these source domains, $\{\mathcal{G}_S^i, \mathbf{Y}_S^i\}_{i=1}^M$, that generalizes well to an unseen target domain $\mathcal{G}_T$ drawn from $P_T(\mathbf{X}, \mathbf{Y}, \mathbf{A})$, where $P_T \neq P_S^i, \forall i \in \{1, \ldots, M\}$. Specifically, for cross-graph node classification with structural shifts (Chen et al., 2025; Wu et al., 2022), we make the following assumptions: the marginal distribution of node features is invariant across domains, i.e., $P_{S^i}(\mathbf{X}) = P_{S^j}(\mathbf{X}) = P(\mathbf{X})$ for any source domains $S^i, S^j$; the conditional distribution of labels given features is invariant, i.e., $P_{S^i}(\mathbf{Y} \mid \mathbf{X}) = P_{S^j}(\mathbf{Y} \mid \mathbf{X}) = P(\mathbf{Y} \mid \mathbf{X})$ for any source domains $S^i, S^j$; and the distribution of the graph structure (adjacency $\mathbf{A}$) depends on the source domain, i.e., $P(\mathbf{A} \mid S^i)$ can change depending on the source domain $S^i$. The objective in cross-domain generalization is to minimize the target risk: $\min_f \mathbb{E}_{(\mathbf{X}_T, \mathbf{Y}_T, \mathbf{A}_T) \sim P_T}[\mathcal{L}_{cls}(f(\mathbf{X}_T, \mathbf{A}_T), \mathbf{Y}_T)]$.

**Graph Attention Networks (GAT).** GAT (Veličković et al., 2018) learns node embeddings by attending over neighbours. For head $k$ at layer $l$,

$$\mathbf{h}_i^{(l+1,k)} = \sigma\Big( \sum_{j \in \mathcal{N}(i) \cup \{i\}} \alpha_{ij}^{(l,k)} \mathbf{W}^{(l,k)} \mathbf{h}_j^{(l)} \Big),$$

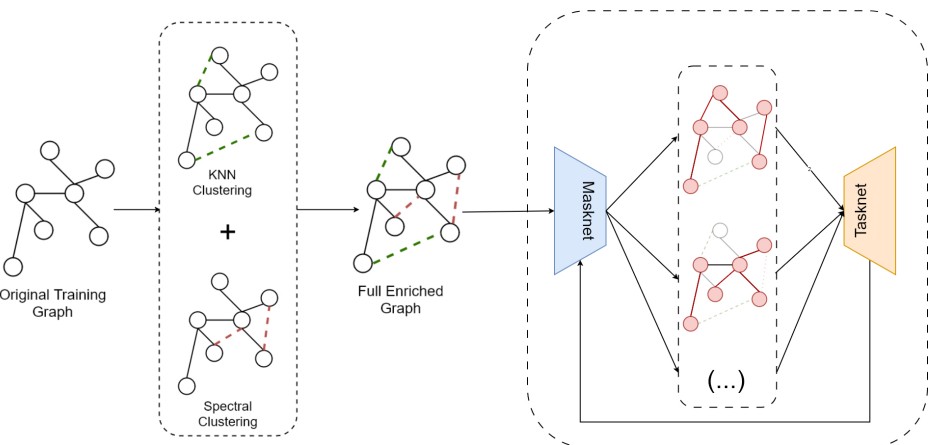

Figure 1: EdgeMask-DG*: (1) enrich the graph with kNN & spectral edges; (2) play a min–max game where MaskNet sparsifies edges and TaskNet (GAT) learns to stay accurate.

where $\alpha_{ij}$ are softmax-normalized attention scores. Outputs from $K$ heads are concatenated. Our model extends GAT by concatenating the learned mask value $s_{ij}$ to the key–query pair, letting low-scoring edges be down-weighted automatically.

**Spectral Clustering.** Spectral clustering (Von Luxburg, 2007) groups data points using the eigenvectors (spectrum) of a Laplacian matrix derived from a similarity graph. For nodes with features $\mathbf{X}$, an affinity matrix $\mathbf{S}$ is built (e.g., using an RBF kernel $S_{ij} = \exp(-\|\mathbf{x}_i - \mathbf{x}_j\|^2/2\zeta^2)$ or cosine similarity). The graph Laplacian (e.g., $\mathbf{L}_{sym} = \mathbf{I} - \mathbf{D}_S^{-1/2}\mathbf{S}\mathbf{D}_S^{-1/2}$) is computed. The eigenvectors corresponding to the $K$ smallest eigenvalues are used to form a $K$-dimensional embedding, which is then clustered. This effectively identifies clusters of nodes that are close in the feature space, providing a basis for feature-derived edges.

**k-Nearest Neighbour (kNN) Graph Construction.** Given a set of node features $\mathbf{X}$, a kNN graph is constructed by creating an edge between a node and its most similar peers. For each node $i$, we identify the set of its $k$ nearest neighbors, $\mathcal{N}_k(i)$, based on a distance metric (e.g., Euclidean distance) or a similarity metric (e.g., cosine similarity) in the feature space $\mathbb{R}^d$. An edge $(i, j)$ is added to the graph if $j \in \mathcal{N}_k(i)$. This method generates a sparse graph structure capturing the most salient local similarities.

## 4 Proposed Approach for Cross-Domain Generalization: EdgeMask-DG*

EdgeMask-DG* aims to learn a GNN $f$ that generalizes across domains with structural shifts by identifying and utilizing domain-invariant substructures. It achieves this through an adversarial learning process operating on an enriched graph structure, using GAT as the backbone. Figure 1 illustrates the two main stages of the EdgeMask-DG* framework: (1) Graph Enrichment augments the source graph topology with feature-derived edges from k-Nearest Neighbors and spectral clustering to form an enriched graph; (2) Adversarial Masking uses a min–max game where MaskNet generates a sparse continuous edge mask to identify challenging perturbations while TaskNet (a GAT) learns to minimize classification loss on the masked graph.

### 4.1 Enriched Graph Representation

Traditional GNNs rely solely on the provided graph structure $\mathcal{E}_S$. However, in Graph-DG with structural shifts, $\mathcal{E}_S$ can contain domain-specific spurious correlations or be insufficient to capture invariant relationships, especially if the source topology is sparse or noisy. Conversely, node features $\mathbf{X}_S$ are assumed to be more stable across domains (Wu et al., 2022). Motivated by this and by prior work such as GraphAug (Chen et al., 2025), EdgeMask-DG* enriches the graph representation by incorporating edges derived from node feature similarity alongside the original topological edges. For each source graph $\mathcal{G}_S = (\mathcal{V}_S, \mathcal{E}_S, \mathbf{X}_S)$, we construct a set of feature-based edges using two complementary strategies. Both the spectral edge set $\mathcal{E}_{Spec}$ and the kNN

edge set $\mathcal{E}_{kNN}$ are generated independently, with each process using the original node features $\mathbf{X}_S$ as input. The strategies are:

To overcome the limitations of domain-specific topology, EdgeMask-DG* constructs an enriched graph by augmenting the original edges $\mathcal{E}_S$ with new edges derived from node feature similarity. We use two complementary strategies. First, we capture global similarity by applying spectral clustering to node features $\mathbf{X}_S$ to identify $K$ communities; edges $\mathcal{E}_{Spec}$ are added between all nodes within the same community. Second, we capture local similarity by building a k-Nearest Neighbors graph based on cosine similarity, creating edges $\mathcal{E}_{kNN}$. To manage complexity, we precompute these full edge sets and, during training, sample subsets $\mathcal{E}_{Spec}^{sampled}$ and $\mathcal{E}_{kNN}^{sampled}$. The final enriched edge set is the union $\mathcal{E}'_S = \mathcal{E}_S \cup \mathcal{E}_{Spec}^{sampled} \cup \mathcal{E}_{kNN}^{sampled}$, which is then used in the adversarial masking process.

The motivation for combining these two distinct feature-based edge generation strategies is to capture complementary aspects of node similarity. Spectral clustering provides a global perspective, ensuring that all nodes within a broad semantic community are connected, which facilitates robust intra-cluster message passing. In contrast, kNN offers a local perspective, creating high-precision connections to a node's most immediate peers. This dual approach creates a more comprehensive and multi-scale structural foundation. The enriched structure $\mathcal{G}'_S = (\mathcal{V}_S, \mathcal{E}'_S, \mathbf{X}_S)$ then serves as the input for our adversarial masking process.

## 4.2   Adversarial Edge Masking

The core of EdgeMask-DG* is a min-max adversarial framework comprising two jointly optimized networks: a TaskNet ($f_\theta$) and a MaskNet ($m_{\phi,\psi}$). The TaskNet is a GAT model, parameterized by $\theta$, that performs node classification. It operates on the node features $\mathbf{X}_S$ and the enriched graph structure $\mathcal{G}'_S$, with its message passing modulated by a continuous edge mask $\mathbf{s}$. The MaskNet is a lightweight network that generates this edge mask. It consists of a feature projection layer $p_\psi : \mathbb{R}^d \to \mathbb{R}^{d'}$ and a mask prediction MLP $g_\phi$. For each edge $(u,v) \in \mathcal{E}'_S$, it computes a score $s_{uv} \in [0,1]$ based on the projected features of the incident nodes:

$$\mathbf{z}_u = \mathrm{ReLU}(p_\psi(\mathbf{x}_u)), \quad \mathbf{z}_v = \mathrm{ReLU}(p_\psi(\mathbf{x}_v)) \quad s_{uv} = \sigma_{\mathrm{sigmoid}}(g_\phi([\mathbf{z}_u, \mathbf{z}_v])) \tag{1}$$

where $[\cdot, \cdot]$ denotes concatenation, $g_\phi$ is a two-layer MLP, and $\sigma_{\mathrm{sigmoid}}$ ensures the output lies in $[0,1]$. The two networks are trained via an alternating optimization procedure to solve a min-max objective over the source domains $\mathcal{D}_S$. In the first step, the TaskNet's parameters $\theta$ are updated to minimize the classification loss, conditioned on the adversarial mask $\mathbf{s}$ generated by a fixed MaskNet. This corresponds to the descent step of the game:

$$\min_\theta \quad \mathbb{E}_{(\mathcal{G}_S, \mathbf{Y}_S) \sim \mathcal{D}_S} [\mathcal{L}_{cls}(f_\theta(\mathbf{X}_S, \mathcal{G}'_S, \mathbf{s}), \mathbf{Y}_S)] \tag{2}$$

During this step, the mask $\mathbf{s} = m_{\phi,\psi}(\mathbf{X}_S, \mathcal{E}'_S)$ is detached from its generator and treated as a fixed edge attribute. Conversely, the MaskNet's parameters $(\phi, \psi)$ are updated to find a mask that maximizes the TaskNet's loss (with fixed $\theta$), while being regularized to enforce sparsity. This ascent step is formulated as an equivalent minimization problem:

$$\min_{\phi,\psi} \quad \mathbb{E}_{(\mathcal{G}_S, \mathbf{Y}_S) \sim \mathcal{D}_S} [-\mathcal{L}_{cls}(f_\theta(\cdot), \mathbf{Y}_S) + \lambda \cdot \mathrm{mean}(m_{\phi,\psi}(\cdot))] \tag{3}$$

where the arguments to the loss and mask functions are as in Eq. 2, $\mathrm{mean}(m_{\phi,\psi}(\dots)) = \frac{1}{|\mathcal{E}'_S|} \sum_{(u,v) \in \mathcal{E}'_S} s_{uv}$, and $\lambda$ is a hyperparameter controlling the sparsity strength. The TaskNet parameters $\theta$ are frozen during this update. This objective encourages the MaskNet to identify edges whose removal or down-weighting most significantly degrades the TaskNet's performance, while the $\lambda$ term prevents it from applying a dense, uninformative mask.

## 4.3   GAT Backbone with Edge Attributes

We assume the enriched graph $\mathcal{G}' = (\mathcal{V}, \mathcal{E}', \mathbf{X}, \mathbf{s})$, where $\mathbf{s} = \{ s_{uv} \in [0,1] \mid (u,v) \in \mathcal{E}' \}$ is the continuous mask produced by the MaskNet. At layer $l$, each node $u \in \mathcal{V}$ has a feature vector $\mathbf{h}_u^{(l)} \in \mathbb{R}^{d_{\mathrm{in}}}$. We employ an $H$-head Graph Attention Network (GAT) in which, for head $k \in \{1, \dots, H\}$, one first computes the linearly

transformed features $\mathbf{z}_u^{(k)} = \mathbf{W}^{(k)} \mathbf{h}_u^{(l)} \in \mathbb{R}^{d_h}$. For each neighbor $v \in \mathcal{N}(u)$, the unnormalized attention logit is then given by

$$e_{uv}^{(k)} = \text{LeakyReLU}\Big(\mathbf{a}^{(k)\top}\big[\mathbf{z}_u^{(k)} \,\|\, \mathbf{z}_v^{(k)} \,\|\, w^{(k)}\, s_{uv}\big]\Big),$$

where $\mathbf{W}^{(k)} \in \mathbb{R}^{d_h \times d_{\text{in}}}$ is a learnable weight matrix for feature transformation, distinct from the learnable scalar weight $w^{(k)} \in \mathbb{R}$ which scales the mask value. The parameters also include the attention vector $\mathbf{a}^{(k)} \in \mathbb{R}^{2d_h+1}$, and $\|$ denotes vector concatenation. The scalar term $w^{(k)}\, s_{uv}$ injects the learned mask into the attention computation, so that edges with low $s_{uv}$ are less likely to receive high attention weights. We normalize these logits via softmax over $v \in \mathcal{N}(u)$ to obtain $\alpha_{uv}^{(k)} = \frac{\exp\big(e_{uv}^{(k)}\big)}{\sum_{w \in \mathcal{N}(u)} \exp\big(e_{uw}^{(k)}\big)}$.

Next, the message passed from $v$ to $u$ under head $k$ is multiplied by $s_{uv}$, yielding $m_{uv}^{(k)} = s_{uv}\, \alpha_{uv}^{(k)}\, \mathbf{z}_v^{(k)}$. In this way, even if $\alpha_{uv}^{(k)}$ is large, a near-zero $s_{uv}$ will completely nullify the contribution of node $v$. Node $u$'s aggregated pre-activation representation for head $k$ is then $\tilde{\mathbf{h}}_u^{(k)} = \sum_{v \in \mathcal{N}(u)} m_{uv}^{(k)}$. At intermediate layers, we apply a nonlinearity $\sigma(\cdot)$ (RELU) and concatenate across heads: $\mathbf{h}_u^{(l+1)} = \big\|_{k=1}^{H} \sigma(\tilde{\mathbf{h}}_u^{(k)}) \in \mathbb{R}^{H\, d_h}$. In the final layer, we instead average across heads before classification: $\mathbf{h}_u^{(L)} = \frac{1}{H} \sum_{k=1}^{H} \sigma(\tilde{\mathbf{h}}_u^{(k)}) \in \mathbb{R}^{d_h}$, and the logits are $\mathbf{o}_u = \mathbf{W}_{\text{out}} \mathbf{h}_u^{(L)}$ with $\mathbf{W}_{\text{out}} \in \mathbb{R}^{C \times d_h}$. When $s_{uv} = 1$ for every edge, this reduces exactly to the standard GAT formulation. When $s_{uv} = 0$, edge $(u, v)$ is entirely suppressed, both in the attention score (since $w^{(k)}\, s_{uv} = 0$) and in the message itself. As a result, the adversarially learned mask $\mathbf{s}$ influences the flow of information during message passing: edges that the MaskNet deems uninformative (low $s_{uv}$) contribute almost nothing to the final node embeddings. This enables the TaskNet to focus on substructures that are robust across domains.

## 4.4 Training Procedure

The overall training process is summarized in Algorithm 1. It is crucial to note that the expectations in the objective functions (Eq. 2 and 3) are taken over the source domain distribution $\mathcal{D}_S$. This is a fundamental constraint of the domain generalization setting, where the target domain is entirely unseen during training. The algorithm, therefore, optimizes the model exclusively on the source graphs. Intuitively, edges that are highly useful only within a particular source graph but not consistently across source domains act as source-specific shortcuts; adversarial pruning suppresses these unstable edges and forces the TaskNet to rely on relations that remain predictive across multiple sources. Because this selection criterion is based solely on cross-source stability under the Graph-DG assumption that $P(\mathbf{X})$ and $P(\mathbf{Y} \mid \mathbf{X})$ are stable while $P(\mathbf{A} \mid S)$ varies, it does not require any information from the unseen target domain.

The procedure involves iterating through epochs, and within each epoch, processing each source domain. For each source graph, we first construct the enriched edge set $\mathcal{E}'_S$. Then, we perform alternating optimization: $N_{descent}$ gradient descent steps are taken on the TaskNet parameters $\theta$ (minimizing Eq. 2), followed by $N_{ascent}$ gradient steps on the MaskNet parameters $\phi, \psi$ (minimizing Eq. 3). This adversarial interplay drives the TaskNet to learn representations that are invariant to the structural perturbations found by the MaskNet, thereby promoting domain generalization.

## 4.5 Theoretical Grounding: Robust Optimization Perspective

The adversarial training procedure of EdgeMask-DG* can be understood through the lens of robust optimization (Ben-Tal et al., 2009; Bertsimas et al., 2011). The MaskNet aims to find a "worst-case" sparse mask $\mathbf{s}$ within a budget $\rho$ (defined by $\frac{1}{m}\|\mathbf{s}\|_1 \leq \rho$) that maximizes the TaskNet's loss $\ell(f_\theta, \mathbf{s}) = \mathcal{L}_{cls}(f_\theta(\mathbf{X}, \mathbf{A}(\mathbf{s})), \mathbf{Y})$. We define this worst-case sparse-mask loss for a fixed $\theta$ as:

$$P(\theta) = \max_{\mathbf{s} \in \mathcal{S}_\rho} \ell(f_\theta, \mathbf{s}), \quad \text{where } \mathcal{S}_\rho = \{\mathbf{s} \in [0,1]^m : \frac{1}{m}\|\mathbf{s}\|_1 \leq \rho\}. \tag{4}$$

**Robust-optimization view.** The problem $P(\theta)$ in equation 4 is upper-bounded by the Lagrangian-penalized objective $D_\lambda(\theta) = \max_{\mathbf{s} \in [0,1]^m}\big[\ell(f_\theta, \mathbf{s}) + \lambda(\frac{1}{m}\|\mathbf{s}\|_1 - \rho)\big]$ for every $\lambda \geq 0$ (weak duality). We therefore train

---

**Algorithm 1** EdgeMask-DG* training procedure

---

1: **Input:** Source graphs $\{(\mathcal{G}_S^i, \mathbf{Y}_S^i)\}_{i=1}^M$, TaskNet $f_\theta$ (GAT), MaskNet $m_{\phi,\psi}$ (Projection $p_\psi$, MLP $g_\phi$), epochs $E$, learning rates $\eta_\theta, \eta_{\phi\psi}$, steps $N_{descent}, N_{ascent}$, sparsity $\lambda$, kNN sample ratio $\gamma_{knn}$, spectral sample ratio $\gamma_{spec}$.
2: Initialize parameters $\theta, \phi, \psi$.
3: Precompute kNN edges $\{\mathcal{E}_{kNN}^i\}_{i=1}^M$ and spectral edges $\{\mathcal{E}_{Spectral}^i\}_{i=1}^M$ for all source graphs.
4: Initialize optimizers $Opt_\theta, Opt_{\phi\psi}$.
5: **for** epoch = 1 to $E$ **do**
6:     **for** each source graph $i \in \{1, \dots, M\}$ **do**
7:         Let $\mathcal{G}_S = \mathcal{G}_S^i = (\mathcal{V}_S, \mathcal{E}_S, \mathbf{X}_S)$, $\mathbf{Y}_S = \mathbf{Y}_S^i$.
8:         Sample $\mathcal{E}_{kNN}^{sampled} \subseteq \mathcal{E}_{kNN}^i$ with ratio $\gamma_{knn}$, and sample $\mathcal{E}_{Spectral}^{sampled} \subseteq \mathcal{E}_{Spectral}^i$ with ratio $\gamma_{spec}$.
9:         Combine edges $\mathcal{E}_S' = \text{coalesce}(\mathcal{E}_S \cup \mathcal{E}_{kNN}^{sampled} \cup \mathcal{E}_{Spectral}^{sampled})$.
10:         // — TaskNet Descent —
11:         **for** $k = 1$ to $N_{descent}$ **do**
12:             Compute mask $\mathbf{s} = m_{\phi,\psi}(\mathbf{X}_S, \mathcal{E}_S')$    // Detach s from $\phi, \psi$
13:             Compute logits $\mathbf{O} = f_\theta(\mathbf{X}_S, \mathcal{E}_S', \mathbf{s})$    // Pass s as edge_attr
14:             Compute loss $\mathcal{L}_{task} = \mathcal{L}_{cls}(\mathbf{O}, \mathbf{Y}_S)$
15:             Update $\theta$ using $\nabla_\theta \mathcal{L}_{task}$ via $Opt_\theta$.
16:         **end for**
17:         // — MaskNet Ascent —
18:         **for** $k = 1$ to $N_{ascent}$ **do**
19:             Compute mask $\mathbf{s} = m_{\phi,\psi}(\mathbf{X}_S, \mathcal{E}_S')$
20:             Compute logits $\mathbf{O} = f_\theta(\mathbf{X}_S, \mathcal{E}_S', \mathbf{s})$    // Detach O from $\theta$
21:             Compute loss $\mathcal{L}_{task} = \mathcal{L}_{cls}(\mathbf{O}, \mathbf{Y}_S)$
22:             Compute regularizer $R(\mathbf{s}) = \text{mean}(\mathbf{s})$
23:             Compute objective $\mathcal{L}_{mask} = -\mathcal{L}_{task} + \lambda R(\mathbf{s})$
24:             Update $\phi, \psi$ using $\nabla_{\phi,\psi} \mathcal{L}_{mask}$ via $Opt_{\phi\psi}$.
25:         **end for**
26:     **end for**
27: **end for**
28: **Output:** Trained TaskNet $f_\theta$.

---

$(\theta, \phi, \psi)$ to find a saddle point for $D_{\lambda_{\text{alg}}}(\theta)$ with a penalty weight $\lambda_{\text{alg}}$. The MaskNet's objective in Algorithm 1 (maximizing $\ell(f_\theta, \mathbf{s}) - \lambda_{\text{alg}} \cdot \text{mean}(\mathbf{s})$) targets the inner maximization $\max_{\mathbf{s}}[\ell(f_\theta, \mathbf{s}) - \lambda_{\text{alg}} \frac{1}{m}\|\mathbf{s}\|_1]$ (ignoring the $-\lambda_{\text{alg}}\rho$ term which is constant w.r.t $\mathbf{s}$ for a fixed $\lambda_{\text{alg}}$). We show in Appendix D that, when the inner maximisation has converged, the resulting masks satisfy the KKT conditions of the constrained problem. In practice, rather than keeping $\lambda_{\text{alg}}$ fixed, one could also update it using a dual ascent step like $\lambda_{\text{alg}} \leftarrow [\lambda_{\text{alg}} + \alpha(\frac{1}{m}\|\mathbf{s}\|_1 - \rho)]_+$, yielding a primal-dual algorithm that can help drive the mean mask value toward $\rho$. For simplicity in this work, we keep $\lambda_{\text{alg}}$ as a fixed hyperparameter.

Furthermore, we can characterize the properties of the optimal adversarial mask $\mathbf{s}^*$ sought by the MaskNet using the Karush-Kuhn-Tucker (KKT) conditions associated with equation 4. The continuous mask values $s_{uv} \in [0, 1]$ generated by our MaskNet (Eq. 1) are differentiable. If employing discrete masks, techniques like Gumbel-softmax (Jang et al., 2016; Maddison et al., 2016) or hard-concrete distributions can provide differentiable relaxations, making the following analysis applicable to such cases.

**Lemma 4.1** (Optimality Conditions for Adversarial Mask)**.** *Let $\mathbf{s}^*$ be an optimal solution to the mask optimization problem equation 4 for a fixed $\theta$. Assuming continuous differentiability of $\ell(f_\theta, \mathbf{s})$ w.r.t. $\mathbf{s}$ and constraint qualifications (Slater's condition holds), there exists an optimal dual variable $\lambda_L^* \geq 0$ (corresponding to the sparsity constraint) such that the optimal mask values $s_e^*$ satisfy the following conditions for all edges $e$.* **(a)** *If $0 < s_e^* < 1$ (edge partially masked):* $\frac{\partial \ell(f_\theta, \mathbf{s}^*)}{\partial s_e} = \frac{\lambda_L^*}{m}$. **(b)** *If $s_e^* = 0$ (edge fully masked):* $\frac{\partial \ell(f_\theta, \mathbf{s}^*)}{\partial s_e} \leq \frac{\lambda_L^*}{m}$. **(c)** *If $s_e^* = 1$ (edge fully included):* $\frac{\partial \ell(f_\theta, \mathbf{s}^*)}{\partial s_e} \geq \frac{\lambda_L^*}{m}$.

**Interpretation:** Lemma 4.1 reveals that the optimal adversarial mask operates based on a threshold mechanism determined by the optimal sparsity cost $\tau^* = \lambda^*/m$. Edges whose marginal contribution to increasing the loss ($\partial\ell/\partial s_e$) exceeds this threshold are fully included ($s_e^* = 1$), edges whose contribution is below the threshold are fully masked ($s_e^* = 0$), and edges whose contribution exactly matches the threshold can be partially masked ($0 < s_e^* < 1$). This provides insight into how the MaskNet prioritizes edges based on their impact on the TaskNet's loss versus the sparsity budget. The alternating training procedure (Algorithm 1) drives the TaskNet to become robust against masks with these properties, encouraging reliance on structures whose removal does not drastically increase the loss.

## 4.6 Computational Complexity

A thorough analysis of the computational complexity for our EdgeMask-DG variants, considering both precomputation and per-epoch training costs, is provided in Appendix B.1. The detailed breakdown incorporates parameters such as the number of nodes ($N$), edges ($M, \tilde{M}$), feature dimensions ($F, P$), GAT architecture ($L, H, h$), and training schedule ($T, D, U, K$). After substituting typical hyperparameter values and focusing on dominant terms, the key findings are as follows: The base EdgeMask-DG model (without additional edges) exhibits an overall complexity of $O(KTN)$, primarily driven by the epoch-wise training on sparse graphs. Introducing kNN augmentation in EdgeMask-DG* (kNN only) shifts the bottleneck to the precomputation phase, resulting in an overall complexity of $O(KN^2)$. The most comprehensive variant, EdgeMask-DG* (kNN + spectral), which also incorporates spectral clustering edges, sees its complexity further increase to $O(KN^3)$, dominated by the spectral precomputation. This progression underscores the trade-off between richer graph representations and computational cost. In particular, the spectral variant is the least scalable component of our framework and is best suited to moderate-scale settings; extending EdgeMask-DG* to larger graphs will require approximate or sparse enrichment schemes.

**Augment-then-prune rationale.** EdgeMask-DG* widens the relational hypothesis space via a union graph of original and feature-guided edges, and then learns a sparse adversarial mask over this union. Importantly, the objective does not encourage indiscriminate removal of edges that are universally useful. Because the TaskNet is trained to maintain low loss under worst-case sparse masking across multiple source domains, edges whose utility is stable and redundant across sources can still be supported by alternative relational paths in the union graph, whereas edges that act as source-specific shortcuts are more likely to be removed. Under the standard Graph-DG assumption that $P(\mathbf{X})$ and $P(\mathbf{Y} \mid \mathbf{X})$ are stable while $P(\mathbf{A} \mid S)$ varies across domains, this pressure favors relational patterns that remain predictive after source-specific edges are suppressed. This argument depends only on source-domain variability and does not require access to the target domain. Empirically, the learned mask prunes both original and augmented edges while retaining 30–39% of augmented edges on average (Table 6), and augmentation helps our method but hurts naive baselines (Table 8).

# 5 Experiments

We evaluate EdgeMask-DG* on a diverse set of benchmarks for node classification under domain shifts, comparing it against various baselines and recent state-of-the-art methods.

## 5.1 Experimental Setup

Our evaluation spans several benchmark datasets prevalent in Graph-DG research, selected to cover diverse distribution shifts, graph properties, and evaluation schemes. A primary set of experiments focuses on the widely used citation networks ACMv9 (A), DBLPv7 (D), and Citationv1 (C).In these datasets, nodes represent academic papers with bag-of-words abstract features (6,775 dimensions) and are classified into 5 research areas. The core challenge arises from significant structural differences across these graphs despite consistent feature and label semantics. To assess broader applicability, we extend our evaluation to additional benchmarks detailed in Table 1. This set includes Cora and Amazon-Photo (featuring artificial transformations), Twitch-explicit, Elliptic, and OGB-Arxiv (characterized by temporal evolution), and Facebook-100 (involving cross-domain transfers). These datasets also vary in their train/validation/test splitting strategies, utilizing

either domain-level or time-aware splits. For the citation networks (ACM, DBLP, Citation), we employ the

Table 1: Summary of datasets. "Artificial Transformation" uses synthetic spurious features to create domain shifts; "Cross-Domain Transfers" uses graphs from distinct domains; "Temporal Evolution" uses dynamic graphs. Splits are "Domain-Level" (by graph) or "Time-Aware" (by time). See Appendix E for details.

| Dataset | #Nodes | #Edges | #Classes | Split | Metric |
|---|---|---|---|---|---|
| *Artificial Transformation* | | | | | |
| Cora | 2,703 | 5,278 | 10 | Domain-Level | Accuracy |
| Amazon-Photo | 7,650 | 119,081 | 10 | Domain-Level | Accuracy |
| *Cross-Domain Transfers* | | | | | |
| Twitch-explicit | 1,912–9,498 | 31,299–153,138 | 2 | Domain-Level | ROC-AUC |
| Facebook-100 | 769–41,536 | 16,656–1,590,655 | 2 | Domain-Level | Accuracy |
| *Temporal Evolution* | | | | | |
| Elliptic | 203,769 | 234,355 | 2 | Time-Aware | F1 Score |
| OGB-Arxiv | 169,343 | 1,166,243 | 40 | Time-Aware | Accuracy |

**Adapted from:** Yang et al. (2016); Shchur et al. (2018); Rozemberczki & Sarkar (2021); Traud et al. (2011); Pareja et al. (2020); Hu et al. (2020).

standard leave-one-domain-out protocol. This involves training a model on two domains (e.g., A and C) and evaluating its performance on the unseen third domain (e.g., D), leading to three distinct scenarios: AC→D, AD→C, and CD→A. Performance is quantified using Micro-F1 and Macro-F1 scores (Sechidis et al., 2011). For the additional datasets (Cora, Photo, FB-100, Twitch, Elliptic, ArXiv), we adhere to their established train/validation/test splits and evaluation metrics—such as Accuracy, ROC-AUC, or F1 score—as specified in Table 1 and common in existing literature.

We benchmark EdgeMask-DG* against a comprehensive set of methods. This includes standard GNNs trained with Empirical Risk Minimization (ERM), such as GCN (Kipf & Welling, 2017), GIN (Xu et al., 2019), GAT (Veličković et al., 2018), and SGC (Wu et al., 2019). We also compare against general domain-generalization and robustness techniques, including ERM, DRNN (Sagawa et al., 2020), MMD (Li et al., 2018), ARM (Zhang et al., 2021), and EERM (Wu et al., 2022). Furthermore, our comparison encompasses specialized Graph-DG methods: EGC (Tailor et al., 2022), ADA (Volpi et al., 2018), MAT (Wang et al., 2022), FLOOD (Liu et al., 2023), MARIO (Zhu et al., 2024), LiSA (Yu et al., 2023), IS-GIB (Yang et al., 2023), and GRM (Wang et al., 2025). Finally, we include recent state-of-the-art approaches like TRACI (Zhao et al., 2025) and GraphAug (Chen et al., 2025), which utilize both GCN and GIN backbones.

EdgeMask-DG* is implemented using PyTorch Geometric (Fey & Lenssen, 2019). The TaskNet ($f_\theta$) is a 4-layer Graph Attention Network (GAT) with 8 attention heads, a hidden dimension of 64 per head, ELU activation, and dropout rates of 0.6 for attention coefficients and 0.5 between layers. The MaskNet ($m_{\phi,\psi}$) comprises a projection layer $p_\psi$ that maps the input node feature dimension $d$ to $d' = 128$, followed by an MLP $g_\phi$ with one hidden layer of size 64. Training employs the Adam optimizer (Kingma & Ba, 2014). For the citation benchmarks (ACM, DBLP, Citation), we use a learning rate of $10^{-3}$, TaskNet weight decay of $5 \times 10^{-4}$, and train for 200 epochs. Key hyperparameters include a sparsity coefficient $\lambda = 10^{-3}$ (for citation benchmarks, potentially varied for others), $N_{descent} = 5$ TaskNet steps, and $N_{ascent} = 1$ MaskNet step per iteration. The enriched graph structure incorporates kNN edges (with $k = 10$ using cosine similarity) and spectral clustering edges (with $K_c = 100$ clusters for citation benchmarks). We sample these feature-derived edges with ratios $\gamma_{knn} = 0.1$ and $\gamma_{spec} = 0.1$, though these parameters are adjusted for different datasets.

## 5.2 Main Results

We present the empirical evaluation of EdgeMask-DG*, comparing its performance against a comprehensive suite of baselines and state-of-the-art methods across various domain generalization benchmarks. The results demonstrate the effectiveness of our proposed adversarial masking on enriched graph structures.

### 5.2.1 Performance on Citation Networks (ACM, DBLP, Citation)

Table 2 summarizes the performance on the cross-graph node classification task using the ACM, DBLP, and Citation datasets. This leave-one-domain-out evaluation directly tests generalization to unseen graph structures. Overall, EdgeMask-DG* achieves the highest total average F1 score of 73.81, surpassing all other methods, including the recent strong baseline GRM (73.28). This indicates the superior overall generalization capability of our approach. Specifically, our method establishes new state-of-the-art performance in two of the three challenging scenarios. In the AD→C scenario, EdgeMask-DG* achieves a Micro-F1 of 79.14% and a Macro-F1 of 77.66%, outperforming all competitors. The improvement in Macro-F1 is particularly notable, suggesting our model's robustness in classifying less frequent classes. Similarly, in the CD→A scenario, our method leads with a Micro-F1 of 71.62% and a Macro-F1 of 72.11%. In the AC→D scenario, while GRM reports the highest scores, EdgeMask-DG* remains highly competitive with a Micro-F1 of 72.54%, performing on par with or better than other recent methods like GraphAug (GIN) and TRACI. These results validate that our proposed mechanism of discovering invariant substructures through adversarial masking on an enriched graph is highly effective for handling the structural shifts inherent in citation networks.

Table 2: Performance on cross-graph node classification (ACM, DBLP, Citation). Results are mean F1 scores (% ± std. dev.). Best results per column are in **bold**. EdgeMask-DG* uses a GAT backbone and adversarial masking on enriched graphs (kNN + Spectral edges).

| Method | AC→D | | AD→C | | CD→A | | Total Avg |
|---|---|---|---|---|---|---|---|
| | Micro F1 | Macro F1 | Micro F1 | Macro F1 | Micro F1 | Macro F1 | |
| MMD | $68.70 \pm 0.88$ | $65.88 \pm 0.54$ | $70.24 \pm 0.77$ | $65.79 \pm 2.40$ | $62.24 \pm 1.48$ | $58.45 \pm 4.15$ | 65.22 |
| ERM | $67.92 \pm 0.50$ | $62.72 \pm 0.58$ | $70.91 \pm 0.61$ | $67.81 \pm 0.69$ | $62.44 \pm 0.98$ | $59.64 \pm 2.34$ | 65.24 |
| DRNN | $69.07 \pm 0.90$ | $65.19 \pm 0.26$ | $71.46 \pm 0.69$ | $67.98 \pm 0.86$ | $63.45 \pm 0.89$ | $60.27 \pm 4.00$ | 66.24 |
| GA (GCN) | $71.45 \pm 0.55$ | $67.19 \pm 0.72$ | $71.62 \pm 0.75$ | $66.09 \pm 0.56$ | $66.44 \pm 0.52$ | $60.70 \pm 0.70$ | 67.25 |
| EERM | $71.31 \pm 0.10$ | $68.39 \pm 0.19$ | $72.91 \pm 0.39$ | $68.47 \pm 1.32$ | $65.28 \pm 1.04$ | $62.98 \pm 2.49$ | 68.22 |
| LiSA | $70.34 \pm 1.45$ | $67.15 \pm 1.45$ | $73.53 \pm 0.60$ | $69.65 \pm 0.88$ | $66.28 \pm 0.67$ | $64.17 \pm 2.43$ | 68.52 |
| TRACI | $72.95 \pm 0.77$ | $69.25 \pm 1.22$ | $74.20 \pm 0.68$ | $69.08 \pm 1.07$ | $67.01 \pm 1.04$ | $65.03 \pm 0.35$ | 69.59 |
| MARIO | $71.71 \pm 0.72$ | $68.66 \pm 0.87$ | $76.05 \pm 0.16$ | $72.38 \pm 0.20$ | $67.90 \pm 0.47$ | $67.13 \pm 0.16$ | 70.64 |
| GA (GIN) | $72.55 \pm 0.60$ | $70.05 \pm 1.71$ | $74.77 \pm 0.60$ | $71.79 \pm 1.18$ | $69.43 \pm 0.65$ | $68.77 \pm 1.07$ | 71.23 |
| GRM | $\mathbf{73.72 \pm 0.75}$ | $\mathbf{71.19 \pm 0.93}$ | $79.13 \pm 0.22$ | $75.74 \pm 0.54$ | $70.23 \pm 1.04$ | $69.66 \pm 2.00$ | 73.28 |
| **EDG\*** | $72.54 \pm 1.41$ | $69.81 \pm 1.76$ | $\mathbf{79.14 \pm 0.19}$ | $\mathbf{77.66 \pm 0.15}$ | $\mathbf{71.62 \pm 0.29}$ | $\mathbf{72.11 \pm 0.25}$ | **73.81** |

### 5.2.2 Performance on Diverse Graph-DG Benchmarks

To assess the broader applicability of EdgeMask-DG*, we evaluated it on datasets exhibiting different types of distribution shifts, as shown in Table 3 and Table 4. On benchmarks with artificial domain shifts (Cora and Photo), EdgeMask-DG* demonstrates exceptional performance, achieving the highest minimum and average accuracy. On Photo, the average accuracy of 94.8% represents a more than 2-percent improvement over the next best method, GRM. The model also performs very strongly on the cross-domain social network Twitch, attaining the highest average ROC-AUC of 59.3%, which is a substantial improvement over prior work.

However, on the FB-100 dataset, our method's performance is less competitive. The average accuracy of 52.1% is lower than that of several baselines, and the minimum accuracy of 45.4 is also comparatively low. This outcome suggests that the core assumption of our method—that node features provide a stable, domain-invariant signal for constructing a reliable enriched graph—may be less effective for the FB-100 dataset, where the topological structure might be overwhelmingly dominant and the node features less informative for generalization.

On the temporal benchmarks, Elliptic and ArXiv, the results presented in Table 4 show varied performance relative to the baselines. While one method, GRM, reports exceptionally high, state-of-the-art scores, these results are unverifiable due to the lack of a public codebase. Among the verifiable methods, EdgeMask-DG* delivers competitive performance. For instance, on Elliptic, its average F1 score of 69.7% is comparable to methods like ARM (69.9%) and MMD (70.6%), though lower than DRNN (71.8%). A similar trend is observed on ArXiv, where our method is competitive with baselines but is outperformed by MARIO and IS-GIB. This suggests that while our approach is robust, specialized methods for handling temporal evolution may have an advantage in these specific settings.

Table 3: OOD Performance on Cora, Photo, FB-100 & Twitch. Min. denotes minimum accuracy/ROC-AUC across domains/runs, Avg. denotes average. Best results are in bold. Values for EdgeMask-DG* are single-seed for Min. (no $\pm$) and include $\pm$ for Avg. where provided.

| Method | Cora | | Photo | | FB-100 | | Twitch | |
|---|---|---|---|---|---|---|---|---|
| | Min. | Avg. | Min. | Avg. | Min. | Avg. | Min. | Avg. |
| ERM | $65.0 \pm 1.5$ | $68.2 \pm 0.4$ | $84.4 \pm 1.5$ | $88.6 \pm 1.3$ | $50.5 \pm 0.4$ | $52.8 \pm 0.6$ | $49.7 \pm 1.1$ | $52.2 \pm 0.9$ |
| DRNN | $56.4 \pm 1.4$ | $74.8 \pm 1.2$ | $76.7 \pm 1.5$ | $77.1 \pm 1.2$ | $48.0 \pm 1.0$ | $51.4 \pm 0.7$ | $44.0 \pm 0.5$ | $48.1 \pm 1.4$ |
| MMD | $52.4 \pm 1.5$ | $75.8 \pm 0.6$ | $82.1 \pm 1.1$ | $84.8 \pm 0.6$ | $51.4 \pm 0.9$ | $53.3 \pm 0.7$ | $42.8 \pm 0.6$ | $49.1 \pm 0.9$ |
| ARM | $60.6 \pm 1.1$ | $62.9 \pm 1.4$ | $58.3 \pm 1.1$ | $74.6 \pm 0.7$ | $50.7 \pm 1.3$ | $54.5 \pm 0.9$ | $43.2 \pm 1.5$ | $48.5 \pm 1.3$ |
| EERM | $68.0 \pm 0.6$ | $70.5 \pm 1.0$ | $90.8 \pm 0.5$ | $91.8 \pm 0.9$ | $50.9 \pm 0.4$ | $54.3 \pm 1.4$ | $51.6 \pm 0.8$ | $54.1 \pm 0.9$ |
| LiSA | $71.1 \pm 1.5$ | $76.7 \pm 0.8$ | $90.3 \pm 1.2$ | $91.5 \pm 1.5$ | $48.8 \pm 1.2$ | $54.2 \pm 1.0$ | $48.6 \pm 1.2$ | $55.8 \pm 2.2$ |
| IS-GIB | $71.3 \pm 1.9$ | $78.6 \pm 1.5$ | $87.2 \pm 0.6$ | $90.2 \pm 0.9$ | $49.6 \pm 1.6$ | $54.6 \pm 1.2$ | $51.2 \pm 1.9$ | $56.0 \pm 1.2$ |
| MARIO | $70.8 \pm 1.3$ | $76.1 \pm 1.0$ | $88.6 \pm 0.8$ | $89.4 \pm 1.4$ | $50.3 \pm 1.9$ | $53.9 \pm 1.4$ | $50.7 \pm 2.0$ | $55.1 \pm 1.9$ |
| GRM | $74.2 \pm 1.2$ | $81.2 \pm 1.5$ | $91.3 \pm 0.9$ | $92.7 \pm 1.6$ | $\mathbf{52.0 \pm 1.3}$ | $\mathbf{55.1 \pm 1.1}$ | $52.5 \pm 1.7$ | $56.7 \pm 1.0$ |
| **EDG*** | $\mathbf{78.0 \pm 1.2}$ | $\mathbf{83.2 \pm 1.1}$ | $\mathbf{94.3 \pm 0.6}$ | $\mathbf{94.8 \pm 0.6}$ | $45.4 \pm 1.2$ | $52.1 \pm 1.8$ | $\mathbf{52.5 \pm 1.3}$ | $\mathbf{59.3 \pm 1.7}$ |

## 6 Ablation Studies

### 6.1 Rationale for Augmentation and Pruning

We now provide empirical justification for EdgeMask-DG*'s two-stage design: first enriching the relational space via graph augmentation, then distilling domain-invariant structures through adversarial pruning.

**Augmentation enriches the structural hypothesis space.** Table 5 demonstrates that incorporating kNN and spectral edges substantially increases the expressiveness of the graph representation across all benchmarks. Augmentation yields a 58.7% increase in edge count for citation networks, accompanied by higher average degree, elevated spectral entropy (indicating greater structural diversity), and reduced feature Laplacian smoothness (reflecting stronger feature–structure coupling). These metrics show that augmentation expands the relational hypothesis space, providing the model with a richer space from which to extract domain-invariant patterns.

**Adversarial masking prunes both sources selectively.** The learned mask does not trivially preserve all augmented edges or remove all original edges; rather, it performs adaptive filtering across both edge types.

Table 4: OOD Performance on Elliptic & ArXiv. T1–T3 denote different time periods. Avg. denotes average. Best results are in bold. *GRM results are unverifiable because the codebase is not publicly available.

| Method | Elliptic | | | | ArXiv | | | |
|---|---|---|---|---|---|---|---|---|
| | T1 | T2 | T3 | Avg. | T1 | T2 | T3 | Avg. |
| ERM | $59.6 \pm 1.4$ | $63.5 \pm 1.3$ | $61.7 \pm 0.6$ | $61.6 \pm 1.1$ | $47.6 \pm 0.9$ | $45.5 \pm 1.4$ | $41.4 \pm 1.0$ | $44.8 \pm 1.4$ |
| EERM | $66.3 \pm 0.4$ | $63.8 \pm 0.6$ | $55.5 \pm 0.6$ | $61.9 \pm 1.1$ | $50.3 \pm 1.4$ | $48.3 \pm 0.4$ | $44.7 \pm 1.4$ | $47.8 \pm 1.4$ |
| ARM | $72.1 \pm 1.5$ | $69.7 \pm 0.7$ | $67.9 \pm 1.4$ | $69.9 \pm 1.3$ | $44.9 \pm 0.7$ | $42.3 \pm 0.6$ | $39.7 \pm 0.8$ | $42.3 \pm 1.0$ |
| LiSA | $68.8 \pm 0.9$ | $65.6 \pm 0.7$ | $69.3 \pm 1.0$ | $67.9 \pm 0.8$ | $45.9 \pm 0.6$ | $42.3 \pm 0.5$ | $46.1 \pm 0.8$ | $44.7 \pm 0.6$ |
| MMD | $71.9 \pm 0.7$ | $70.1 \pm 0.4$ | $69.9 \pm 0.8$ | $70.6 \pm 0.8$ | $44.6 \pm 1.3$ | $42.4 \pm 0.7$ | $38.9 \pm 1.0$ | $42.0 \pm 0.5$ |
| **EDG*** | $70.7 \pm 1.2$ | $68.9 \pm 0.9$ | $68.2 \pm 1.3$ | $69.7 \pm 1.0$ | $48.1 \pm 1.1$ | $45.8 \pm 0.8$ | $44.2 \pm 1.2$ | $45.9 \pm 1.0$ |
| DRNN | $73.2 \pm 1.4$ | $71.4 \pm 0.7$ | $70.6 \pm 0.3$ | $71.8 \pm 0.8$ | $46.8 \pm 0.5$ | $44.7 \pm 1.1$ | $40.5 \pm 1.3$ | $44.0 \pm 1.0$ |
| IS-GIB | $71.2 \pm 1.1$ | $70.0 \pm 1.0$ | $70.4 \pm 1.2$ | $70.5 \pm 1.1$ | $49.3 \pm 0.8$ | $46.6 \pm 0.9$ | $50.5 \pm 1.3$ | $48.8 \pm 0.7$ |
| MARIO | $69.8 \pm 1.9$ | $72.8 \pm 2.4$ | $71.1 \pm 1.4$ | $71.2 \pm 2.0$ | $48.8 \pm 2.3$ | $50.1 \pm 2.4$ | $49.2 \pm 2.4$ | $49.4 \pm 2.8$ |
| *GRM* * | $\mathbf{89.4 \pm 1.5}$ | $\mathbf{85.5 \pm 1.1}$ | $\mathbf{89.1 \pm 1.5}$ | $\mathbf{88.0 \pm 1.4}$ | $\mathbf{52.2 \pm 0.9}$ | $\mathbf{52.6 \pm 1.4}$ | $\mathbf{56.1 \pm 1.4}$ | $\mathbf{53.6 \pm 1.2}$ |

Table 5: Subspace enrichment after augmentation (5 seeds; mean $\pm$ std)

| Dataset / Split | $|E|$ increase (%) | Avg. degree $\uparrow$ | Spectral entropy $\uparrow$ | Feat. Laplacian smoothness $\downarrow$ |
|---|---|---|---|---|
| ACM/DBLP/Citation (avg.) | $58.7 \pm 2.9$ | $2.1 \pm 0.1$ | $0.26 \pm 0.02$ | $0.14 \pm 0.01$ |
| Cora | $47.5 \pm 3.2$ | $1.6 \pm 0.1$ | $0.19 \pm 0.02$ | $0.11 \pm 0.02$ |
| Amazon-Photo | $38.4 \pm 2.8$ | $1.4 \pm 0.1$ | $0.17 \pm 0.02$ | $0.09 \pm 0.02$ |
| Twitch (avg.) | $41.2 \pm 3.1$ | $1.7 \pm 0.2$ | $0.18 \pm 0.03$ | $0.10 \pm 0.02$ |
| Facebook-100 (avg.) | $36.9 \pm 2.4$ | $1.3 \pm 0.2$ | $0.15 \pm 0.02$ | $0.07 \pm 0.01$ |
| Elliptic (T1–T3 avg.) | $33.1 \pm 2.1$ | $0.9 \pm 0.1$ | $0.12 \pm 0.02$ | $0.06 \pm 0.01$ |
| OGB-Arxiv (T1–T3 avg.) | $29.8 \pm 2.0$ | $0.8 \pm 0.1$ | $0.10 \pm 0.02$ | $0.05 \pm 0.01$ |

As shown in Table 6, the adversarial mechanism prunes approximately 68.3% of augmented edges while simultaneously removing 18.9% of original edges, yet retains 30–39% of augmented edges on average. From this we can infer that augmented edges contribute non-trivial domain-invariant information that complements, rather than replaces information in the original topology.

Table 6: Mask behavior: fraction of edges pruned/retained (5 seeds; mean $\pm$ std).

| Dataset / Split | Pruned from Aug (%) | Pruned from Orig (%) | Aug edges retained (%) |
|---|---|---|---|
| Citation avg. | $68.3 \pm 2.1$ | $18.9 \pm 1.5$ | $31.7 \pm 2.1$ |
| Cora | $70.5 \pm 2.4$ | $16.2 \pm 1.3$ | $29.5 \pm 2.4$ |
| Amazon-Photo | $65.4 \pm 2.2$ | $21.7 \pm 1.6$ | $34.6 \pm 2.2$ |
| Twitch (avg.) | $66.8 \pm 2.7$ | $20.9 \pm 1.8$ | $33.2 \pm 2.7$ |
| Facebook-100 avg. | $64.2 \pm 2.5$ | $22.5 \pm 1.7$ | $35.8 \pm 2.5$ |
| Elliptic avg. | $61.3 \pm 2.0$ | $24.1 \pm 1.5$ | $38.7 \pm 2.0$ |
| OGB-Arxiv avg. | $62.7 \pm 2.1$ | $23.4 \pm 1.6$ | $37.3 \pm 2.1$ |

**Combined effect**  A $2 \times 2$ study (Table 7) confirms that both augmentation and masking are necessary, and their combination is synergistic. Training with masking on the original graph alone (Orig + Mask) provides modest gains over standard EERM. Training on the augmented graph without masking (Union / No-Mask) yields further improvement. However, the full EdgeMask-DG* method (Union + Mask) consistently achieves the highest performance, with improvements ranging from 0.5 to 2.3 percentage points over the next-best variant across all benchmarks.

Table 7: $2 \times 2$ ablation (5 seeds; mean $\pm$ std). "No-Mask" = EERM, "Mask" = adversarial masking.

| Dataset / Metric | Orig / No-Mask | **Orig + Mask** | Union / No-Mask | **Union + Mask (ours)** |
|---|---|---|---|---|
| AC→D (Micro-F1 %) | $69.5 \pm 0.7$ | $70.8 \pm 0.6$ | $71.6 \pm 0.6$ | $72.54 \pm 1.41$ |
| AD→C (Micro-F1 %) | $75.5 \pm 0.6$ | $77.0 \pm 0.5$ | $78.6 \pm 0.4$ | $79.14 \pm 0.19$ |
| CD→A (Micro-F1 %) | $66.8 \pm 0.7$ | $68.2 \pm 0.6$ | $70.5 \pm 0.5$ | $71.62 \pm 0.29$ |
| Cora (Acc %) | $81.0 \pm 0.9$ | $82.1 \pm 0.8$ | $82.6 \pm 0.8$ | $83.2 \pm 1.1$ |
| Amazon-Photo (Acc %) | $92.5 \pm 0.6$ | $93.7 \pm 0.5$ | $94.3 \pm 0.5$ | $94.8 \pm 0.6$ |
| Twitch (ROC-AUC) | $55.2 \pm 1.2$ | $57.0 \pm 1.1$ | $58.6 \pm 1.0$ | $59.3 \pm 1.7$ |
| Facebook-100 (Acc %) | $50.1 \pm 1.2$ | $51.2 \pm 1.1$ | $51.8 \pm 1.1$ | $52.1 \pm 1.8$ |
| Elliptic (F1 %) | $67.2 \pm 1.1$ | $68.6 \pm 1.0$ | $69.2 \pm 0.9$ | $69.7 \pm 1.0$ |
| OGB-Arxiv (Acc %) | $44.3 \pm 1.0$ | $45.1 \pm 0.9$ | $45.6 \pm 0.9$ | $45.9 \pm 1.0$ |

**Unique benefit of adversarial filtering.**  Table 8 reveals a distinction: naive augmentation (training directly on the union graph) degrades performance for standard baselines (ERM, EERM, GRM, IS-GIB), with $\Delta$ values ranging from $-0.9$ to $-0.2$ across datasets. In contrast, EdgeMask-DG* consistently benefits from augmentation, achieving positive $\Delta$ values of $+0.8$ to $+2.3$ percentage points. This demonstrates that the adversarial masking mechanism is essential. Without it, the added edges introduce more noise than signal, whereas with it, the model successfully identifies and leverages domain-invariant augmented structures.

Table 8: $\Delta$ from augmentation (Augmented minus Original). Mean $\pm$ std over 5 seeds.

| Dataset / Split | ERM $\Delta$ | EERM $\Delta$ | GRM $\Delta$ | IS-GIB $\Delta$ | **Ours $\Delta$** |
|---|---|---|---|---|---|
| Cora | $-0.8 \pm 0.6$ | $-0.6 \pm 0.5$ | $-0.5 \pm 0.6$ | $-0.2 \pm 0.5$ | $+1.1 \pm 0.5$ |
| Amazon-Photo | $-0.5 \pm 0.5$ | $-0.7 \pm 0.5$ | $-0.3 \pm 0.5$ | $-0.3 \pm 0.4$ | $+1.1 \pm 0.4$ |
| Twitch | $-0.9 \pm 0.7$ | $-1.1 \pm 0.6$ | $-0.6 \pm 0.6$ | $-0.5 \pm 0.6$ | $+2.3 \pm 0.6$ |
| Facebook-100 | $-0.7 \pm 0.6$ | $-1.0 \pm 0.6$ | $-0.4 \pm 0.6$ | $-0.3 \pm 0.5$ | $+0.9 \pm 0.5$ |
| Elliptic | $-0.4 \pm 0.5$ | $-0.5 \pm 0.5$ | $-0.2 \pm 0.5$ | $-0.2 \pm 0.4$ | $+1.1 \pm 0.5$ |
| OGB-Arxiv | $-0.6 \pm 0.5$ | $-0.8 \pm 0.5$ | $-0.3 \pm 0.5$ | $-0.2 \pm 0.4$ | $+0.8 \pm 0.3$ |

### 6.1.1  Effect of Sparsity Regularization $\lambda$

We evaluate EdgeMask-DG* with varying values of the sparsity coefficient $\lambda$ in Eq. 3. The results for the CD→A scenario (on citation networks) are shown in Table 9. The results in Table 9 demonstrate that the sparsity regularization term is a key component for effective generalization. While the model performs well without any explicit sparsity penalty ($\lambda = 0$), introducing a non-zero $\lambda$ consistently improves performance. The model achieves optimal or near-optimal results across a range of small values, particularly between $10^{-5}$ and $10^{-2}$, indicating robustness to the exact choice of this hyperparameter within this effective range. The explicit sparsity term helps guide the MaskNet to find challenging yet meaningful perturbations, preventing it from simply masking a large number of edges.

Table 9: Ablation on the sparsity coefficient ($\lambda$) for the CD$\rightarrow$A scenario (5 seeds; mean $\pm$ std). Metrics are F1 scores (%).

| $\lambda$ | Micro-F1 (%) | Macro-F1 (%) |
|---|---|---|
| 0.0 | $70.60 \pm 0.35$ | $71.54 \pm 0.38$ |
| 1e$-$5 | $\mathbf{71.78 \pm 0.28}$ | $\mathbf{72.69 \pm 0.30}$ |
| 1e$-$4 | $70.85 \pm 0.42$ | $71.48 \pm 0.44$ |
| 1e$-$3 | $71.59 \pm 0.26$ | $72.18 \pm 0.29$ |
| 1e$-$2 | $71.63 \pm 0.27$ | $72.65 \pm 0.31$ |
| 1e$-$1 | $71.49 \pm 0.30$ | $72.18 \pm 0.33$ |

## 7    Conclusion

We introduced EdgeMask-DG*, a novel Graph-DG framework that learns domain-invariant substructures to handle structural shifts. It operates via a min-max game on an enriched graph, where an adversarial masker challenges a GAT-based classifier by finding worst-case sparse edge masks. This forces the classifier to become robust to structural perturbations and rely on generalizable patterns. Our approach, which combines original topology with feature-derived edges, overcomes the limitations of static augmentations. Experiments show that EdgeMask-DG* attains the best average results on the citation benchmark and strong, competitive performance across several additional settings, while the ablations confirm the value of both the enriched graph and the adversarial mechanism. Future work should focus on more scalable graph-enrichment procedures, especially approximate alternatives to spectral augmentation, and on extensions to graph-level tasks.

## Acknowledgements

Naresh Manwani gratefully acknowledges support from ANRF under grant CRG/2023/007970.

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

# A Appendix

## A.1 Dataset Statistics

Detailed statistics for the citation network datasets used in our experiments are provided in Table 10.

Table 10: Statistics of the citation network datasets used for domain generalization.

| Dataset | Abbrev. | # Nodes | # Edges | # Features | # Classes |
|---------|---------|---------|---------|------------|-----------|
| ACMv9 | A | 9,360 | 15,602 | 6,775 | 5 |
| Citationv1 | C | 8,935 | 15,113 | 6,775 | 5 |
| DBLPv7 | D | 5,484 | 8,130 | 6,775 | 5 |

# B Reproducibility Statement

All datasets used in this study are publicly available and cited appropriately. Specifically, the citation network datasets (ACMv9, DBLPv7, Citationv1) follow the splits and preprocessing from prior works (Chen et al., 2025; Wu et al., 2022). Other benchmark datasets like Cora, Amazon-Photo, Twitch-explicit, Facebook-100, Elliptic, and OGB-Arxiv are standard benchmarks with established splits detailed in Section 5 and Table 1, with references to their original sources.

Our model, EdgeMask-DG*, is implemented in PyTorch using PyTorch Geometric. Key architectural details (GAT: 4 layers, 8 heads, 64 dim/head; MaskNet: projection to 128 dim, MLP with one 64-dim hidden layer) and training hyperparameters (Adam optimizer, LR $10^{-3}$ for citation nets, WD $5 \times 10^{-4}$, 200 epochs, $\lambda = 10^{-3}$ sparsity, $N_{descent} = 5, N_{ascent} = 1$) are provided in Section 5 and Appendix B. Specifics for feature edge generation (kNN: k=10, cosine; Spectral: $K_c = 100$ clusters, RBF kernel, sample ratios $\gamma = 0.1$) are also detailed. Ablation studies in Appendix F further explore hyperparameter sensitivity.

The source code for our experiments, including scripts for data loading, model training, and evaluation, is provided at the anonymized URL: https://github.com/rbSparky/TMLR. This repository contains instructions to reproduce all main results and ablation studies presented. The computational experiments were run on NVIDIA 2080 GPUs.

### B.1 Time Complexity Analysis

We now provide a time-complexity analysis for the three proposed variants of our EdgeMask-DG framework. Throughout this analysis, we consistently use the following notation: $N = |V|$ represents the number of nodes, $M = |E|$ is the count of original edges, and $F$ denotes the input feature dimension. The mask-projection dimension is $P$, while $H$ signifies the hidden size per GAT head, with $h$ attention heads and $L$ GAT layers. The training regimen includes $D$ TaskNet ("descent") steps and $U$ MaskNet ("ascent") steps per epoch. This process is repeated for $K$ source-domain graphs over a total of $T$ training epochs. Additional edges introduced by kNN and spectral clustering are denoted $E_{\text{kNN}}$ and $E_{\text{spec}}$, respectively. The total number of edges in the graph used for computation is $\tilde{M} = M + E_{\text{kNN}} + E_{\text{spec}}$.

Each training epoch executes $D + U$ inner loops, iterating over the $K$ source graphs. Within each such inner loop, the computational costs are primarily composed of four components. First, the mask projection step incurs a cost of $O(N \cdot F \cdot P)$. Second, the MaskNet forward pass requires $O(\tilde{M} \cdot P)$. Third, the TaskNet forward and backward passes, which involve a stack of GATConv layers, cost $O(L \cdot \tilde{M} \cdot (h \cdot H))$. Finally, computing losses and penalties (cross-entropy and sparsity regularization) takes $O(N + \tilde{M})$. Summing these, and noting that the $O(N + \tilde{M})$ term is dominated by others (assuming $P \geq 1$ and $LhH \geq 1$), one inner step has an asymptotic cost of

$$O\big(NFP \;+\; \tilde{M}P \;+\; L\tilde{M}hH\big).$$

Consequently, a full epoch, encompassing all $K$ graphs and $D + U$ inner loops, costs

$$O\big((D + U)\,K\,(NFP + \tilde{M}P + L\tilde{M}hH)\big).$$

### 1. EdgeMask-DG (base, no extra edges)

For the base EdgeMask-DG variant, no additional edges are precomputed or added, so $\tilde{M} = M$. The precomputation cost is therefore zero. The per-epoch complexity is $O\big((D + U)\,K\,(NFP + MP + LMhH)\big)$. Over $T$ training epochs, the overall complexity is

$$\boxed{O\big(T\,(D + U)\,K\,(NFP + MP + LMhH)\big).}$$

### 2. EdgeMask-DG* (kNN only)

In the EdgeMask-DG* variant augmented solely with kNN edges, we first precompute the kNN graph for each of the $K$ domains. Using a standard implementation like kneighbors_graph, this takes $O(N^2)$ per graph, leading to a total precomputation cost of $O(KN^2)$. These $E_{\text{kNN}}$ edges are then added to the original $M$ edges, so $\tilde{M} = M + E_{\text{kNN}}$. The per-epoch training cost becomes $O\big((D+U)\,K\,(NFP+(M+E_{\text{kNN}})P+L(M+E_{\text{kNN}})hH)\big)$. The overall complexity is the sum of precomputation and total training costs:

$$\boxed{O\big(K\,N^2 \;+\; T\,(D + U)\,K\,(NFP + (M + E_{\text{kNN}})P + L(M + E_{\text{kNN}})hH)\big).}$$

### 3. EdgeMask-DG* (kNN + spectral)

When both kNN and spectral edges are incorporated, the precomputation phase is more intensive. For each graph, spectral edge generation involves constructing an affinity matrix and performing eigen-decomposition, costing $O(N^3)$, followed by edge sampling, which is typically $O(N^2)$. Thus, spectral precomputation is $O(N^3)$ per graph. The kNN precomputation remains $O(N^2)$ per graph. Across $K$ graphs, the total precomputation cost is $O(K(N^3 + N^2))$, which simplifies to $O(KN^3)$. With these additional $E_{\text{spec}}$ and $E_{\text{kNN}}$ edges, the total edge count becomes $\tilde{M} = M + E_{\text{kNN}} + E_{\text{spec}}$. The per-epoch training cost is $O\big((D + U)\,K\,(NFP + \tilde{M}P + L\tilde{M}hH)\big)$. The overall complexity is therefore

$$\boxed{O\big(K\,N^3 \;+\; T\,(D + U)\,K\,(NFP + (M + E_{\text{kNN}} + E_{\text{spec}})P + L(M + E_{\text{kNN}} + E_{\text{spec}})hH)\big).}$$

**Worst-case Simplification**

Under worst-case assumptions where added kNN and spectral edges result in dense graphs, i.e., $E_{\text{kNN}}, E_{\text{spec}} = O(N^2)$, the total edge count $\tilde{M}$ becomes $O(N^2)$. In this scenario, the dominant term in the per-epoch complexity for all variants, when summed over $T$ epochs, is $O(T(D+U)KLhHN^2)$. For the EdgeMask-DG* (kNN + spectral) variant, the precomputation cost is $O(KN^3)$. Comparing these, the heaviest scenario across all variants and phases is dictated by the spectral precomputation and the dense-graph training, leading to an overall worst-case complexity of

$$\boxed{O\!\left(K\,N^3 \;+\; T\,(D+U)\,K\,L\,hH\,N^2\right).}$$

**Simplification with Default Hyperparameters**

We now simplify the complexity expressions by substituting default hyperparameter values commonly used in our experiments and then dropping negligible terms.

**Common Starting Point for Simplification**    The cost for one inner loop over a single graph, including all four components (Mask projection, MaskNet forward, TaskNet forward/backward, and Losses/penalties), is $NFP + \tilde{M}P + L\tilde{M}hH + N + \tilde{M}$. The total training cost over $T$ epochs, $D + U$ inner loops per epoch (where $D = 5$ descent and $U = 1$ ascent steps, so $D + U = 6$), and $K = 3$ source graphs, is

$$\text{train-cost} = T(D+U)K\big[NFP + \tilde{M}P + L\tilde{M}hH + N + \tilde{M}\big].$$

We use the default constants from our implementation: $F = 7, P = 128, H = 128, h = 8, L = 4$, and $T = 200$. Let $C_1 = (D+U)K = 6 \times 3 = 18$. The term $FP + 1$ becomes $(7 \times 128) + 1 = 896 + 1 = 897$. The term $P + LhH + 1$ becomes $128 + (4 \times 8 \times 128) + 1 = 128 + 4096 + 1 = 4225$. Substituting these into the training cost formula yields:

$$\text{train-cost} = TC_1\big[N(FP+1) + \tilde{M}(P + LhH + 1)\big] = 200 \times 18\big[N(897) + \tilde{M}(4225)\big].$$

The leading numeric factor $200 \times 18 \times 897 \approx 3.23 \times 10^6$ for the $N$-term and $200 \times 18 \times 4225 \approx 1.52 \times 10^7$ for the $\tilde{M}$-term are constant multipliers. Asymptotically, only the terms involving $N$ and $\tilde{M}$ (and other variables like $T, K$) matter, so the training cost is broadly $O(TK(N + \tilde{M}))$.

**1. EdgeMask-DG (no extra edges)**    For the base variant, $\tilde{M} = M$. In sparse graphs like social networks, $M \approx cN$ for a small constant average degree $c$ (e.g., $20 - 30$). The general training cost $O\big(TK(N + \tilde{M})\big)$, upon substituting $\tilde{M} \sim O(N)$, simplifies to $O\big(TK(N + N)\big) = O\big(TKN\big)$. Since there is no pre-computation cost, this is the overall complexity.

$$\boxed{\textbf{EdgeMask-DG: } O\big(KTN\big)}$$

This simplified form depends on the number of nodes $N$, the number of domains $K$, and the number of epochs $T$.

**2. EdgeMask-DG* (kNN edges only)**    The pre-computation for kNN graphs (e.g., $k = 10$) is $O(KN^2)$. During training, kNN adds $E_{\text{kNN}} = kN$ edges (assuming undirected edges, this is a constant degree addition). Thus, $\tilde{M} = M + E_{\text{kNN}} \approx (c + k)N = \Theta(N)$ if $M \approx cN$. The training cost therefore remains $O(TKN)$. To compare magnitudes, for a graph with $N = 10,000$, and with $T = 200, K = 3$: The pre-computation cost $KN^2 = 3 \times (10^4)^2 = 3 \times 10^8$. The training cost $KTN = 3 \times 200 \times 10^4 = 6 \times 10^6$. The quadratic pre-computation cost clearly dominates the training cost.

$$\boxed{\textbf{EdgeMask-DG* (kNN): } O\big(KN^2\big)}$$

The dominant variables are $N$ and $K$.

**3. EdgeMask-DG\* (kNN + spectral)** The pre-computation for spectral clustering is $O(N^3)$ per graph (dominated by eigen-decomposition), leading to a total pre-computation cost of $O(KN^3)$. Spectral clustering can add $E_{\text{spec}} = \rho N^2$ edges (e.g., with default $\rho = 0.1$), making the graph dense. Thus, $\tilde{M} = M + E_{\text{kNN}} + E_{\text{spec}} = O(N) + O(N) + O(N^2) = \Theta(N^2)$. The training cost $O\big(TK(N + \tilde{M})\big)$ becomes $O\big(TK(N + N^2)\big) = O(TKN^2)$. Comparing magnitudes with $N = 10,000, T = 200, K = 3$: The pre-computation cost $KN^3 = 3 \times (10^4)^3 = 3 \times 10^{12}$. The training cost $KTN^2 = 3 \times 200 \times (10^4)^2 = 6 \times 10^{10}$. The cubic pre-computation cost is again the bottleneck by a significant margin.

$$\boxed{\textbf{EdgeMask-DG* (kNN + spectral): } O\big(KN^3\big)}$$

The complexity is primarily determined by $N$ and $K$.

## C    Theoretical Analysis: Robust Optimization Perspective

We provide a theoretical foundation for EdgeMask-DG\* by framing its adversarial training as solving a robust optimization problem over edge masks. This perspective highlights how the method encourages robustness to structural perturbations.

### C.1    Problem Setup

Let $\mathcal{G}' = (\mathcal{V}, \mathcal{E}', \mathbf{X})$ be an enriched graph instance with $N = |\mathcal{V}|$ nodes and $m = |\mathcal{E}'|$ edges. Let $\mathbf{Y}$ be the node labels. The TaskNet $f_\theta$ maps the graph features $\mathbf{X}$ and an adjacency structure $\mathbf{A}(\mathbf{s})$ modulated by an edge mask $\mathbf{s} \in [0,1]^m$ to node logits. The average cross-entropy loss on this graph is denoted by:

$$\ell(f_\theta, \mathbf{s}) := \frac{1}{N} \sum_{v \in \mathcal{V}} \mathcal{L}_{\text{CE}}\Big([f_\theta(\mathbf{X}, \mathbf{A}(\mathbf{s}))]_v, \; [\mathbf{Y}]_v\Big).$$

We define an uncertainty set $\mathcal{S}_\rho$ for the masks based on an $\ell_1$ sparsity budget $\rho \in (0,1]$:

$$\mathcal{S}_\rho := \Big\{ \mathbf{s} \in [0,1]^m \; : \; \frac{1}{m}\|\mathbf{s}\|_1 \leq \rho \Big\}, \quad \text{where } \|\mathbf{s}\|_1 = \sum_{e=1}^{m} s_e.$$

Consider the robust optimization problem of finding the mask $\mathbf{s} \in \mathcal{S}_\rho$ that maximizes the TaskNet's loss for a fixed $\theta$:

$$P(\theta) := \max_{\mathbf{s} \in \mathcal{S}_\rho} \ell(f_\theta, \mathbf{s}). \tag{P'}$$

The goal of robust training is to find parameters $\theta$ that minimize this worst-case loss: $\min_\theta P(\theta)$.

### C.2    Lagrangian Duality and Penalized Objective

We demonstrate that the constrained maximization problem equation P' is equivalent to an unconstrained maximization problem with an $\ell_1$ penalty term, achieved through Lagrangian duality.

**Lemma C.1** (Lagrangian Upper Bound). *For any Lagrange multiplier $\lambda_L \geq 0$ and any fixed $\theta$, the worst-case loss $P(\theta) = \max_{\mathbf{s} \in \mathcal{S}_\rho} \ell(f_\theta, \mathbf{s})$ is upper-bounded by the dual function $D(\lambda_L; \theta)$:*

$$P(\theta) \leq D(\lambda_L; \theta) := \max_{\mathbf{s} \in [0,1]^m} \left[ \ell(f_\theta, \mathbf{s}) + \lambda_L \left( \frac{1}{m}\|\mathbf{s}\|_1 - \rho \right) \right]. \tag{5}$$

*Consequently, $P(\theta) \leq \min_{\lambda_L \geq 0} D(\lambda_L; \theta)$. Equality, $P(\theta) = \min_{\lambda_L \geq 0} D(\lambda_L; \theta)$, holds (i.e., strong duality) if $\ell(f_\theta, \mathbf{s})$ is concave in $\mathbf{s}$ and other regularity conditions are met. However, since $\ell(f_\theta, \mathbf{s})$ (which involves GNN computations and cross-entropy loss) is generally non-concave in $\mathbf{s}$, we only have weak duality (the upper bound).*

*Proof.* The primal problem is $P(\theta) = \max_{\mathbf{s}} \ell(f_\theta, \mathbf{s})$ subject to $\mathbf{s} \in [0,1]^m$ and $g(\mathbf{s}) = \frac{1}{m}\|\mathbf{s}\|_1 - \rho \le 0$.

The Lagrangian is $\mathcal{L}(\mathbf{s}, \lambda_L; \theta) = \ell(f_\theta, \mathbf{s}) + \lambda_L g(\mathbf{s}) = \ell(f_\theta, \mathbf{s}) + \lambda_L \left(\frac{1}{m}\|\mathbf{s}\|_1 - \rho\right)$, for $\lambda_L \ge 0$.

The dual function is $D(\lambda_L; \theta) = \max_{\mathbf{s} \in [0,1]^m} \mathcal{L}(\mathbf{s}, \lambda_L; \theta)$. Since $\ell(f_\theta, \mathbf{s})$ is continuous and $[0,1]^m$ is compact, the maximum is attained.

For any $\mathbf{s}$ feasible for the primal problem (i.e., $\mathbf{s} \in \mathcal{S}_\rho$, so $g(\mathbf{s}) \le 0$) and any $\lambda_L \ge 0$:

$$\ell(f_\theta, \mathbf{s}) \le \ell(f_\theta, \mathbf{s}) - \lambda_L g(\mathbf{s})\% \text{ since } \lambda_L \ge 0 \text{ and } g(\mathbf{s}) \le 0 \implies -\lambda_L g(\mathbf{s}) \ge 0$$

The term $-\lambda_L g(\mathbf{s})$ is non-negative. More accurately, $\ell(f_\theta, \mathbf{s}) \le \mathcal{L}(\mathbf{s}, \lambda_L; \theta)$ because $\lambda_L g(\mathbf{s}) \le 0$. This seems off for maximization. Let's restate for standard weak duality in maximization. For any $\mathbf{s}' \in \mathcal{S}_\rho$ (feasible for primal) and any $\lambda_L \ge 0$:

$$\ell(f_\theta, \mathbf{s}') \le \ell(f_\theta, \mathbf{s}') - \lambda_L \left(\frac{1}{m}\|\mathbf{s}'\|_1 - \rho\right)$$

because $\lambda_L \ge 0$ and $\left(\frac{1}{m}\|\mathbf{s}'\|_1 - \rho\right) \le 0$. Then,

$$\ell(f_\theta, \mathbf{s}') \le \max_{\mathbf{s} \in [0,1]^m} \left[\ell(f_\theta, \mathbf{s}) + \lambda_L \left(\frac{1}{m}\|\mathbf{s}\|_1 - \rho\right)\right] = D(\lambda_L; \theta).$$

Since this holds for any feasible $\mathbf{s}'$, it holds for the $\mathbf{s}^*$ that maximizes $\ell(f_\theta, \mathbf{s}')$ over $\mathcal{S}_\rho$. Thus, $P(\theta) = \max_{\mathbf{s} \in \mathcal{S}_\rho} \ell(f_\theta, \mathbf{s}) \le D(\lambda_L; \theta)$ for any $\lambda_L \ge 0$. This implies $P(\theta) \le \min_{\lambda_L \ge 0} D(\lambda_L; \theta)$, which is the statement of weak duality.

Strong duality, $P(\theta) = \min_{\lambda_L \ge 0} D(\lambda_L; \theta)$, holds if the primal problem is a convex optimization problem (i.e., maximizing a concave function over a convex set) and satisfies constraint qualifications (e.g., Slater's condition). In our case, $\ell(f_\theta, \mathbf{s})$ is generally non-concave due to the non-linear GNN and the convex (not concave) nature of the cross-entropy loss with respect to logits, which themselves are non-linear functions of $\mathbf{s}$. Therefore, strong duality is not guaranteed. The saddle-point formulation presented previously relied on strong duality and thus may not hold in the general non-concave case. Thus, to show the intuition behind the proposed method, we show this analysis on a concave surrogate as well. $\square$

## C.3 Interpretation and Connection to Algorithm 1

Lemma C.1 establishes that the dual function $D(\lambda_L; \theta)$ provides an upper bound on the true robust objective $P(\theta)$. Our algorithm aims to find parameters $(\theta, \phi, \psi)$ that optimize a related penalized objective.

Algorithm 1 implements an alternating gradient descent-ascent procedure. For a fixed sparsity hyperparameter $\lambda_{\text{alg}} \ge 0$ (this is the $\lambda$ used in Eq. 3 and Algorithm 1), the algorithm seeks parameters $(\theta^*, \phi^*, \psi^*)$ such that:

- $\theta^*$ minimizes $\ell(f_\theta, \mathbf{s}^*)$ given $\mathbf{s}^* = m_{\phi^*, \psi^*}(\cdot)$.

- $(\phi^*, \psi^*)$ minimizes $-\ell(f_{\theta^*}, \mathbf{s}) + \lambda_{\text{alg}} \cdot \text{mean}(\mathbf{s})$ over masks $\mathbf{s}$ generated by $m_{\phi, \psi}(\cdot)$. This is equivalent to maximizing $\ell(f_{\theta^*}, \mathbf{s}) - \lambda_{\text{alg}} \cdot \text{mean}(\mathbf{s})$.

The MaskNet's objective (maximization form: $\ell(f_\theta, \mathbf{s}) - \lambda_{\text{alg}} \cdot \text{mean}(\mathbf{s})$) targets the inner maximization of a Lagrangian-like expression $\max_{\mathbf{s}}[\ell(f_\theta, \mathbf{s}) - \frac{\lambda_{\text{alg}}}{m}\|\mathbf{s}\|_1]$. This corresponds to the term $\ell(f_\theta, \mathbf{s}) + \lambda_L(\frac{1}{m}\|\mathbf{s}\|_1 - \rho)$ from $D(\lambda_L; \theta)$ if we associate $\lambda_{\text{alg}}/m$ with $-\lambda_L/m$ (for maximization of loss) or $\lambda_L/m$ (for minimization of negative loss), and note that the term $-\lambda_L \rho$ is constant with respect to $\mathbf{s}$ for a fixed $\lambda_L$ and is thus dropped during the MaskNet's optimization of $\mathbf{s}$.

## C.4 Analysis of a Concave Surrogate Inner Game (s is discrete add relaxation)

While the true loss function $\ell(f_\theta, \mathbf{s})$ is generally non-concave in $\mathbf{s}$, we can gain further insight by analyzing a simplified surrogate problem where the inner objective is concave. This allows us to establish strong duality and derive a closed-form optimal mask, providing an intuition for the behavior of the MaskNet.

Consider an affine approximation of the loss function $\ell(f_\theta, \mathbf{s})$ linearized around $\mathbf{s} = \mathbf{0}$:

$$\tilde{\ell}(f_\theta, \mathbf{s}) = \ell(f_\theta, \mathbf{0}) + \nabla_{\mathbf{s}}\ell(f_\theta, \mathbf{0})^\top \mathbf{s}. \tag{6}$$

This surrogate loss $\tilde{\ell}(f_\theta, \mathbf{s})$ is affine in $\mathbf{s}$, and therefore also concave in $\mathbf{s}$. Let $c_e(\theta) = \nabla_{s_e}\ell(f_\theta, \mathbf{0})$ be the gradient of the original loss with respect to the $e$-th mask component, evaluated at $\mathbf{s} = \mathbf{0}$. Then $\tilde{\ell}(f_\theta, \mathbf{s}) = \ell(f_\theta, \mathbf{0}) + \sum_{e=1}^{m} c_e(\theta)s_e$.

### C.4.1 Optimal Mask for the Penalized Surrogate Problem

We first analyze the optimal mask for a penalized version of the surrogate problem, which directly relates to the objective optimized by the MaskNet in Algorithm 1 if it were using this surrogate loss. The MaskNet's update rule (Eq. 3) aims to maximize $\mathcal{L}_{cls}(f_\theta, \mathbf{s}) - \lambda \cdot \text{mean}(\mathbf{s})$. Using the surrogate loss $\tilde{\ell}$ and defining the penalty strength $\tau = \lambda_{\text{alg}}/m$ (where $\lambda_{\text{alg}}$ is the sparsity hyperparameter from Algorithm 1), the objective becomes:

$$\max_{\mathbf{s} \in [0,1]^m} \left[ \tilde{\ell}(f_\theta, \mathbf{s}) - \tau \sum_{e=1}^{m} s_e \right]. \tag{7}$$

Substituting Eq. equation 6:

$$\max_{\mathbf{s} \in [0,1]^m} \left[ \ell(f_\theta, \mathbf{0}) + \sum_{e=1}^{m} c_e(\theta)s_e - \tau \sum_{e=1}^{m} s_e \right]$$

$$= \ell(f_\theta, \mathbf{0}) + \max_{\mathbf{s} \in [0,1]^m} \sum_{e=1}^{m} (c_e(\theta) - \tau) \, s_e.$$

This is a linear program in $\mathbf{s}$, where each $s_e$ is constrained to $[0, 1]$ and can be chosen independently to maximize its term in the sum. For each edge $e$:

- If $c_e(\theta) - \tau > 0$, the term $(c_e(\theta) - \tau)s_e$ is maximized by setting $s_e^* = 1$.

- If $c_e(\theta) - \tau < 0$, the term $(c_e(\theta) - \tau)s_e$ is maximized by setting $s_e^* = 0$.

- If $c_e(\theta) - \tau = 0$, any $s_e \in [0, 1]$ is optimal for that term; we can choose $s_e^* = 0$ by convention (or $s_e^* = 1$).

Thus, the optimal mask $\mathbf{s}^*$ for the penalized surrogate problem has components:

**Proposition C.2** (Optimal Mask for Penalized Surrogate). *The optimal mask $\mathbf{s}^* = (s_1^*, \ldots, s_m^*)$ that solves the penalized surrogate problem defined in equation 7 is given by:*

$$s_e^* = \mathbb{I}\left[\nabla_{s_e}\ell(f_\theta, \mathbf{0}) > \tau\right], \tag{8}$$

*where $\tau = \lambda_{alg}/m$ is the effective sparsity threshold and $\mathbb{I}[\cdot]$ is the indicator function (equal to 1 if the condition is true, 0 otherwise).*

*Proof.* As derived above, maximizing $\sum_{e=1}^{m}(c_e(\theta) - \tau)s_e$ subject to $s_e \in [0, 1]$ involves setting $s_e = 1$ if its coefficient $(c_e(\theta) - \tau)$ is positive, and $s_e = 0$ if its coefficient is negative. This directly leads to $s_e^* = \mathbb{I}[c_e(\theta) > \tau]$. $\square$

### C.4.2 Strong Duality for the Constrained Surrogate Problem

Now, consider the original constrained robust optimization problem $P(\theta)$ from Eq. equation 4 (or equation P'), but using the surrogate loss $\tilde{\ell}$:

$$P_{surr}(\theta) := \max_{\mathbf{s} \in \mathcal{S}_\rho} \tilde{\ell}(f_\theta, \mathbf{s}), \quad \text{where } \mathcal{S}_\rho = \left\{ \mathbf{s} \in [0,1]^m \; : \; \frac{1}{m}\|\mathbf{s}\|_1 \le \rho \right\}. \tag{9}$$

Since $\tilde{\ell}(f_\theta, \mathbf{s})$ is affine (and thus concave) in $\mathbf{s}$, and the feasible set $\mathcal{S}_\rho$ is convex and compact, strong duality holds for this problem (e.g., Slater's condition is satisfied if $\rho > 0$ by choosing $\mathbf{s} = \mathbf{0}$ as a strictly feasible point). Therefore, $P_{surr}(\theta) = \min_{\lambda_D \geq 0} D_{surr}(\lambda_D; \theta)$, where $D_{surr}(\lambda_D; \theta)$ is the dual function:

$$
\begin{aligned}
D_{surr}(\lambda_D; \theta) &= \max_{\mathbf{s} \in [0,1]^m} \left[ \tilde{\ell}(f_\theta, \mathbf{s}) - \lambda_D \left( \frac{1}{m} \sum_{e=1}^m s_e - \rho \right) \right] \\
&= \max_{\mathbf{s} \in [0,1]^m} \left[ \ell(f_\theta, \mathbf{0}) + \sum_{e=1}^m c_e(\theta) s_e - \frac{\lambda_D}{m} \sum_{e=1}^m s_e + \lambda_D \rho \right] \\
&= \ell(f_\theta, \mathbf{0}) + \lambda_D \rho + \max_{\mathbf{s} \in [0,1]^m} \sum_{e=1}^m \left( c_e(\theta) - \frac{\lambda_D}{m} \right) s_e.
\end{aligned}
\tag{10}
$$

Let $\tau_D = \lambda_D/m$. The inner maximization $\max_{\mathbf{s} \in [0,1]^m} \sum_{e=1}^m (c_e(\theta) - \tau_D) s_e$ is solved by $s_e^*(\lambda_D) = \mathbb{I}[c_e(\theta) > \tau_D]$. The optimal $\mathbf{s}^{**}$ for the constrained problem $P_{surr}(\theta)$ will be $s_e^{**} = \mathbb{I}[c_e(\theta) > \lambda_D^*/m]$, where $\lambda_D^*$ is the optimal dual variable that minimizes $D_{surr}(\lambda_D; \theta)$. This $\lambda_D^*$ is chosen such that the resulting mask $\mathbf{s}^{**}$ satisfies the budget constraint $\frac{1}{m} \sum s_e^{**} = \rho$ (if the constraint is active) or $\frac{1}{m} \sum s_e^{**} < \rho$ (if $\lambda_D^* = 0$).

### C.4.3 Connection to the EdgeMask-DG* Algorithm

The EdgeMask-DG* algorithm does not directly use the surrogate loss $\tilde{\ell}$ based on $\nabla_{\mathbf{s}} \ell(f_\theta, \mathbf{0})$. Instead, its MaskNet uses the gradients of the true loss $\ell(f_\theta, \mathbf{s})$ evaluated at the current mask $\mathbf{s}_{curr}$, i.e., $\nabla_{\mathbf{s}} \ell(f_\theta, \mathbf{s}_{curr})$, to guide its updates. However, the analysis of the surrogate problem provides a valuable intuition:

- It demonstrates that if the loss landscape were simple (affine), the optimal strategy for the MaskNet (given a fixed sparsity penalty $\tau$) would be a hard thresholding of edges based on their gradient scores $c_e(\theta)$. Edges whose inclusion contributes to increasing the loss by more than $\tau$ are kept ($s_e = 1$), while others are discarded ($s_e = 0$).

- The EdgeMask-DG* algorithm, through its adversarial training and the MaskNet's objective (Eq. 3), can be viewed as attempting to learn such a thresholding behavior, but adapted to the complex, non-linear landscape of the true loss $\ell(f_\theta, \mathbf{s})$. The MaskNet computes edge scores $s_{uv}$ (Eq. 1) based on node features and aims to find masks that are impactful yet sparse.

- The gradient ascent performed by the MaskNet on $-\mathcal{L}_{cls} + \lambda \cdot \text{mean}(\mathbf{s})$ iteratively seeks regions where a sparse mask maximally degrades TaskNet performance. The structure of the optimal mask for the surrogate (Proposition C.2) suggests that such masks would prioritize edges with high "scores" (sensitivity of loss) relative to the sparsity cost.

Therefore, while the true optimization is more complex, the concave surrogate analysis reinforces the idea that the MaskNet learns to identify and exploit a critical subset of edges, akin to a learned thresholding mechanism based on edge importance and sparsity budget. This iterative process, acting on the true non-concave loss but sharing similarities with the optimal strategy for the concave surrogate, drives the TaskNet to become robust to the removal/down-weighting of edges deemed less critical or potentially spurious by this adaptive thresholding. It is important to reiterate that strong duality and the closed-form solution of Proposition C.2 apply to the simplified concave surrogate problem. Hence, these specific results apply only to the surrogate, not directly to the original non-concave game, for which we only have weak duality.

## D KKT Analysis of the Adversarial Mask Optimization

We analyze the properties of the optimal mask $\mathbf{s}^*$ that the MaskNet implicitly seeks for a fixed TaskNet $f_\theta$. This corresponds to solving the inner maximization problem from the robust optimization perspective (Problem equation P' in Section C):

$$
\max_{\mathbf{s} \in \mathbb{R}^m} \quad \ell(f_\theta, \mathbf{s})
\tag{11}
$$

subject to:

$$g_1(\mathbf{s}) = \frac{1}{m}\sum_{e=1}^{m} s_e - \rho \le 0 \tag{12}$$

$$h_e(\mathbf{s}) = s_e - 1 \le 0 \quad \forall e \in \{1,\ldots,m\} \tag{13}$$

$$k_e(\mathbf{s}) = -s_e \le 0 \quad \forall e \in \{1,\ldots,m\} \tag{14}$$

where $\ell(f_\theta, \mathbf{s})$ is the TaskNet loss, $m = |\mathcal{E}'|$, and $\rho \in (0,1]$ is the sparsity budget. We assume $\ell(f_\theta, \mathbf{s})$ is continuously differentiable with respect to $\mathbf{s}$.

To analyze the solution $\mathbf{s}^*$, we formulate the Karush-Kuhn-Tucker (KKT) conditions. It is standard to formulate KKT conditions for minimization problems. We consider the equivalent minimization problem:

$$\min_{\mathbf{s}\in\mathbb{R}^m} \quad -\ell(f_\theta, \mathbf{s})$$

subject to the same constraints equation 12, equation 13, equation 14.

Let $\lambda \ge 0$ be the Lagrange multiplier for the sparsity constraint $g_1(\mathbf{s}) \le 0$. Let $\mu_e \ge 0$ be the Lagrange multipliers for the upper bound constraints $h_e(\mathbf{s}) \le 0$. Let $\nu_e \ge 0$ be the Lagrange multipliers for the lower bound constraints $k_e(\mathbf{s}) \le 0$.

The Lagrangian function is:

$$\mathcal{L}(\mathbf{s}, \lambda, \boldsymbol{\mu}, \boldsymbol{\nu}) = -\ell(f_\theta, \mathbf{s}) + \lambda g_1(\mathbf{s}) + \sum_{e=1}^{m}\mu_e h_e(\mathbf{s}) + \sum_{e=1}^{m}\nu_e k_e(\mathbf{s})$$

$$= -\ell(f_\theta, \mathbf{s}) + \lambda\left(\frac{1}{m}\sum_{e=1}^{m}s_e - \rho\right) + \sum_{e=1}^{m}\mu_e(s_e - 1) + \sum_{e=1}^{m}\nu_e(-s_e)$$

The KKT conditions for an optimal solution $\mathbf{s}^*$ (assuming constraint qualifications like Slater's condition hold, which is true here as $\mathbf{s} = \mathbf{0}$ is strictly feasible for $g_1$ if $\rho > 0$) are:

1. **Stationarity:** The gradient of the Lagrangian with respect to $\mathbf{s}$ must be zero at $\mathbf{s}^*$:

$$\nabla_{\mathbf{s}}\mathcal{L}(\mathbf{s}^*, \lambda^*, \boldsymbol{\mu}^*, \boldsymbol{\nu}^*) = \mathbf{0}$$

For each component $s_e$, this gives:

$$-\frac{\partial\ell(f_\theta, \mathbf{s}^*)}{\partial s_e} + \lambda^*\left(\frac{1}{m}\right) + \mu_e^*(1) + \nu_e^*(-1) = 0$$

Rearranging:

$$\frac{\partial\ell(f_\theta, \mathbf{s}^*)}{\partial s_e} = \frac{\lambda^*}{m} + \mu_e^* - \nu_e^* \quad \forall e \in \{1,\ldots,m\} \tag{15}$$

2. **Primal Feasibility:** The solution $\mathbf{s}^*$ must satisfy all original constraints:

$$\frac{1}{m}\sum_{e=1}^{m}s_e^* - \rho \le 0 \tag{16}$$

$$s_e^* - 1 \le 0 \quad \forall e \tag{17}$$

$$-s_e^* \le 0 \quad \forall e \tag{18}$$

3. **Dual Feasibility:** All Lagrange multipliers must be non-negative:

$$\lambda^* \ge 0, \quad \mu_e^* \ge 0, \quad \nu_e^* \ge 0 \quad \forall e \tag{19}$$

4. **Complementary Slackness:** The product of each multiplier and its corresponding constraint value must be zero at $\mathbf{s}^*$:

$$\lambda^* \left( \frac{1}{m} \sum_{e=1}^{m} s_e^* - \rho \right) = 0 \tag{20}$$

$$\mu_e^*(s_e^* - 1) = 0 \quad \forall e \tag{21}$$

$$\nu_e^*(-s_e^*) = 0 \quad \Leftrightarrow \quad \nu_e^* s_e^* = 0 \quad \forall e \tag{22}$$

It is important to note that because the inner maximization problem equation 11 (maximizing $\ell(f_\theta, \mathbf{s})$) is generally non-convex with respect to $\mathbf{s}$ (due to the non-linear GNN operations), the KKT conditions characterize necessary conditions for local optima or stationary points. They do not guarantee that a point satisfying them is the global maximizer of the MaskNet's objective. During the adversarial training (Algorithm 1), the MaskNet's optimization step (ascent) aims to find such a point, typically approximating a local maximum but the above analysis provides a theoretical justification for the algorithm's objective.

### D.1 Interpretation of KKT Conditions

We analyze the stationarity condition equation 15 based on the optimal value $s_e^*$ using the complementary slackness conditions equation 21 and equation 22. Let $g_e^* = \frac{\partial \ell(f_\theta, \mathbf{s}^*)}{\partial s_e}$ denote the gradient of the loss with respect to the mask value of edge $e$ at the optimum.

- **Case 1: Interior solution** $(0 < s_e^* < 1)$
  From equation 22, since $s_e^* > 0$, we must have $\nu_e^* = 0$.
  From equation 21, since $s_e^* < 1$ (i.e., $s_e^* - 1 < 0$), we must have $\mu_e^* = 0$.
  Substituting $\mu_e^* = 0$ and $\nu_e^* = 0$ into the stationarity condition equation 15:

$$g_e^* = \frac{\partial \ell(f_\theta, \mathbf{s}^*)}{\partial s_e} = \frac{\lambda^*}{m}$$

  *Interpretation:* For edges that are partially masked, the marginal increase in loss obtained by slightly increasing the mask value $s_e$ must exactly equal the marginal "cost" imposed by the sparsity constraint, represented by $\lambda^*/m$.

- **Case 2: Fully masked edge** $(s_e^* = 0)$
  From equation 21, $\mu_e^*(0 - 1) = -\mu_e^* = 0$, which implies $\mu_e^* = 0$.
  Condition equation 22 $(\nu_e^* s_e^* = 0)$ is automatically satisfied, and we only know $\nu_e^* \geq 0$.
  Substituting $\mu_e^* = 0$ into the stationarity condition equation 15:

$$g_e^* = \frac{\partial \ell(f_\theta, \mathbf{s}^*)}{\partial s_e} = \frac{\lambda^*}{m} - \nu_e^*$$

  Since $\nu_e^* \geq 0$, this implies:

$$\frac{\partial \ell(f_\theta, \mathbf{s}^*)}{\partial s_e} \leq \frac{\lambda^*}{m}$$

  *Interpretation:* An edge is completely removed $(s_e^* = 0)$ if the marginal gain in loss from slightly increasing its mask value is less than or equal to the marginal sparsity cost $\lambda^*/m$. Including this edge even partially is not "worth" the sparsity cost it incurs.

- **Case 3: Fully included edge** $(s_e^* = 1)$
  From equation 22, $\nu_e^*(1) = 0$, which implies $\nu_e^* = 0$.
  Condition equation 21 $(\mu_e^*(s_e^* - 1) = 0)$ is automatically satisfied, and we only know $\mu_e^* \geq 0$.
  Substituting $\nu_e^* = 0$ into the stationarity condition equation 15:

$$g_e^* = \frac{\partial \ell(f_\theta, \mathbf{s}^*)}{\partial s_e} = \frac{\lambda^*}{m} + \mu_e^*$$

Since $\mu_e^* \geq 0$, this implies:

$$\frac{\partial \ell(f_\theta, \mathbf{s}^*)}{\partial s_e} \geq \frac{\lambda^*}{m}$$

*Interpretation:* An edge is fully kept ($s_e^* = 1$) if the marginal gain in loss from increasing its mask value (evaluated at $s_e = 1$) is greater than or equal to the marginal sparsity cost $\lambda^*/m$. The contribution of this edge to maximizing the loss outweighs its sparsity cost.

Furthermore, the complementary slackness condition equation 20 tells us about the optimal Lagrange multiplier $\lambda^*$ for the sparsity constraint:

- If the sparsity constraint is inactive at the optimum (i.e., $\frac{1}{m} \sum s_e^* < \rho$), then we must have $\lambda^* = 0$. In this scenario, the sparsity budget is not limiting, and the optimal mask $\mathbf{s}^*$ is determined solely by maximizing $\ell(f_\theta, \mathbf{s})$ within the box $[0,1]^m$. The conditions simplify: $g_e^* = \mu_e^* - \nu_e^*$.

- If the sparsity constraint is active at the optimum (i.e., $\frac{1}{m} \sum s_e^* = \rho$), then $\lambda^*$ can be positive ($\lambda^* \geq 0$). In this case, $\lambda^*/m$ acts as a non-zero threshold on the loss gradient $\frac{\partial \ell}{\partial s_e}$ required to include an edge ($s_e^* > 0$). A higher $\lambda^*$ (resulting from a tighter budget $\rho$) imposes a stricter requirement for including edges.

# E  Gradient Analysis of the Adversarial Game

We analyze the gradient dynamics of the alternating optimization procedure used in Algorithm 1 to understand how the interplay between the TaskNet ($f_\theta$) and MaskNet ($m_{\phi,\psi}$) encourages robustness. Let $\mathcal{L}_{cls}(\theta, \phi, \psi)$ denote the classification loss for a given graph instance, making the parameter dependencies explicit:

$$\mathcal{L}_{cls}(\theta, \phi, \psi) = \mathcal{L}_{cls}(f_\theta(\mathbf{X}, \mathcal{G}', \mathbf{s}), \mathbf{Y}), \quad \text{where } \mathbf{s} = m_{\phi,\psi}(\mathbf{X}, \mathcal{E}').$$

Let $R(\phi, \psi)$ denote the sparsity regularization term:

$$R(\phi, \psi) = \lambda \cdot \text{mean}(\mathbf{s}) = \frac{\lambda}{m} \|\mathbf{s}\|_1 = \frac{\lambda}{m} \sum_{e=1}^{m} s_e, \quad \text{where } \mathbf{s} = m_{\phi,\psi}(\mathbf{X}, \mathcal{E}').$$

## E.1  TaskNet Gradient Update

The TaskNet aims to minimize the classification loss given the current fixed mask $\mathbf{s}$ generated by the MaskNet. The objective for the TaskNet update step is:

$$J_{task}(\theta) = \mathcal{L}_{cls}(\theta, \phi_{\text{fixed}}, \psi_{\text{fixed}})$$

The gradient used for updating the TaskNet parameters $\theta$ is:

$$\mathbf{g}_\theta = \nabla_\theta J_{task}(\theta) = \nabla_\theta \mathcal{L}_{cls}(f_\theta(\mathbf{X}, \mathcal{G}', \mathbf{s}), \mathbf{Y}) \tag{23}$$

where $\mathbf{s} = m_{\phi_{\text{fixed}}, \psi_{\text{fixed}}}(\cdot)$ is treated as a constant input (edge attributes) during this computation. This is a standard gradient computation for the GAT model $f_\theta$ operating on the graph $\mathcal{G}'$ with edge attributes $\mathbf{s}$. The update rule is $\theta \leftarrow \theta - \eta_\theta \mathbf{g}_\theta$. This step adapts the TaskNet to perform well on the graph structure as perturbed by the current adversarial mask.

## E.2  MaskNet Gradient Update

The MaskNet aims to find mask parameters $(\phi, \psi)$ that maximize the TaskNet's loss (for fixed $\theta$) while minimizing the average mask value (promoting sparsity). This is formulated as minimizing the following objective:

$$J_{mask}(\phi, \psi) = -\mathcal{L}_{cls}(\theta_{\text{fixed}}, \phi, \psi) + R(\phi, \psi)$$

The gradient used for updating the MaskNet parameters $(\phi, \psi)$ is $\mathbf{g}_{\phi,\psi} = \nabla_{\phi,\psi} J_{mask}(\phi, \psi)$. We compute this gradient using the chain rule. Let $\mathbf{p} = (\phi, \psi)$ denote the combined MaskNet parameters.

$$\mathbf{g}_{\phi,\psi} = \nabla_{\mathbf{p}} J_{mask}(\phi, \psi) = \nabla_{\mathbf{p}}[-\mathcal{L}_{cls}(\theta_{\text{fixed}}, \phi, \psi)] + \nabla_{\mathbf{p}}[R(\phi, \psi)]$$

**Term 1: Gradient of Negative Loss** The loss $\mathcal{L}_{cls}$ depends on $\mathbf{p}$ through the mask $\mathbf{s} = m_{\mathbf{p}}(\cdot)$.

$$\nabla_{\mathbf{p}}[-\mathcal{L}_{cls}] = -\nabla_{\mathbf{p}}[\mathcal{L}_{cls}]$$

Applying the chain rule:

$$\nabla_{\mathbf{p}}[\mathcal{L}_{cls}] = \frac{\partial \mathcal{L}_{cls}}{\partial \mathbf{s}} \frac{\partial \mathbf{s}}{\partial \mathbf{p}}$$

Here:

- $\frac{\partial \mathcal{L}_{cls}}{\partial \mathbf{s}} = \nabla_{\mathbf{s}} \mathcal{L}_{cls}$ is the gradient of the TaskNet's loss with respect to the mask vector $\mathbf{s}$. This is a row vector of size $1 \times m$. Its $e$-th element, $\frac{\partial \mathcal{L}_{cls}}{\partial s_e}$, represents the sensitivity of the loss to the mask value of edge $e$.

- $\frac{\partial \mathbf{s}}{\partial \mathbf{p}} = \nabla_{\mathbf{p}} \mathbf{s}$ is the Jacobian matrix of the MaskNet output $\mathbf{s}$ with respect to its parameters $\mathbf{p}$. Its size is $m \times |\mathbf{p}|$. The $(e, j)$-th element is $\frac{\partial s_e}{\partial p_j}$.

The product is a row vector of size $1 \times |\mathbf{p}|$, representing the gradient of the loss with respect to the MaskNet parameters.

$$\nabla_{\mathbf{p}}[-\mathcal{L}_{cls}] = -(\nabla_{\mathbf{s}} \mathcal{L}_{cls})(\nabla_{\mathbf{p}} \mathbf{s})$$

**Term 2: Gradient of Sparsity Regularizer**

$$R(\phi, \psi) = \frac{\lambda}{m} \sum_{e=1}^{m} s_e(\mathbf{p})$$

$$\nabla_{\mathbf{p}}[R(\phi, \psi)] = \frac{\lambda}{m} \nabla_{\mathbf{p}} \left[ \sum_{e=1}^{m} s_e(\mathbf{p}) \right] = \frac{\lambda}{m} \sum_{e=1}^{m} \nabla_{\mathbf{p}}[s_e(\mathbf{p})]$$

Using Jacobian notation:

$$\sum_{e=1}^{m} \nabla_{\mathbf{p}}[s_e(\mathbf{p})] = \mathbf{1}^T (\nabla_{\mathbf{p}} \mathbf{s})$$

where $\mathbf{1}$ is an $m \times 1$ column vector of ones. So,

$$\nabla_{\mathbf{p}}[R(\phi, \psi)] = \frac{\lambda}{m} \mathbf{1}^T (\nabla_{\mathbf{p}} \mathbf{s})$$

**Combined MaskNet Gradient:** Substituting the terms back:

$$\begin{aligned}
\mathbf{g}_{\phi,\psi} &= \nabla_{\mathbf{p}} J_{mask}(\phi, \psi) \\
&= -(\nabla_{\mathbf{s}} \mathcal{L}_{cls})(\nabla_{\mathbf{p}} \mathbf{s}) + \frac{\lambda}{m} \mathbf{1}^T (\nabla_{\mathbf{p}} \mathbf{s}) \\
&= \left( -\nabla_{\mathbf{s}} \mathcal{L}_{cls} + \frac{\lambda}{m} \mathbf{1}^T \right) (\nabla_{\mathbf{p}} \mathbf{s})
\end{aligned} \tag{24}$$

The update rule for the MaskNet is $\mathbf{p} \leftarrow \mathbf{p} - \eta_{\phi\psi} \mathbf{g}_{\phi,\psi}^T$ (transposing the row gradient to a column vector for parameter updates).

### E.3 Interpretation of Gradient Dynamics

The gradient update for the MaskNet parameters $\mathbf{p} = (\phi, \psi)$ is driven by two main components, filtered through the MaskNet's Jacobian $\nabla_{\mathbf{p}}\mathbf{s}$:

1. **Loss Sensitivity ($-\nabla_{\mathbf{s}}\mathcal{L}_{cls}$):** This term pushes the MaskNet parameters to change the mask $\mathbf{s}$ in a direction that increases the TaskNet loss $\mathcal{L}_{cls}$. Specifically, if $\frac{\partial \mathcal{L}_{cls}}{\partial s_e}$ is positive (increasing $s_e$ increases loss), the gradient term $-\frac{\partial \mathcal{L}_{cls}}{\partial s_e}$ is negative, encouraging changes in $\mathbf{p}$ that lead to a decrease in $s_e$. Conversely, if $\frac{\partial \mathcal{L}_{cls}}{\partial s_e}$ is negative (increasing $s_e$ decreases loss), this term encourages an increase in $s_e$. The MaskNet learns to identify edges where the TaskNet is sensitive.

2. **Sparsity Pressure ($\frac{\lambda}{m}\mathbf{1}^T$):** This term provides a constant positive pressure in the objective gradient (or negative pressure in the parameter update direction $-\mathbf{g}_{\phi,\psi}$) for all mask values $s_e$, scaled by $\lambda/m$. It encourages the MaskNet parameters to change in ways that decrease all $s_e$ values, promoting overall sparsity.

The MaskNet parameters $\mathbf{p} = (\phi, \psi)$ are updated to minimize $J_{mask}(\phi, \psi) = -\mathcal{L}_{cls}(\theta_{\text{fixed}}, \phi, \psi) + R(\phi, \psi)$. The gradient $\mathbf{g}_{\phi,\psi}$ is given by Equation equation 24. Since Algorithm 1 employs an optimizer that minimizes $J_{mask}$, the parameter update is $\mathbf{p} \leftarrow \mathbf{p} - \eta_{\phi\psi}\mathbf{g}_{\phi,\psi}^T$.

To understand how this update affects individual mask values $s_e$, consider the term $Q_e = -\frac{\partial \mathcal{L}_{cls}}{\partial s_e} + \frac{\lambda}{m}$, which is the $e$-th component of the vector $\left(-\nabla_{\mathbf{s}}\mathcal{L}_{cls} + \frac{\lambda}{m}\mathbf{1}^T\right)$ in Equation equation 24. The change in $s_e$ is driven by this $Q_e$ through the MaskNet's Jacobian $\nabla_{\mathbf{p}}\mathbf{s}$. Specifically, because the MaskNet uses a sigmoid output for $s_e$ (Equation 1), the entries of the Jacobian term $\frac{\partial s_e}{\partial(\text{pre-sigmoid activation for } s_e)}$ are non-negative. Assuming the parameters $\mathbf{p}$ primarily influence $s_e$ through its pre-sigmoid activation in a way that $\frac{\partial s_e}{\partial \mathbf{p}}$ components (relevant to $s_e$) don't invert the sign of $Q_e$'s influence, the effective change $\Delta s_e$ will have a sign opposite to $Q_e$. Thus, the MaskNet will tend to:

- **Increase** $s_e$ if $Q_e < 0$, which means $\frac{\partial \mathcal{L}_{cls}}{\partial s_e} > \frac{\lambda}{m}$ (the loss increase from $s_e$ outweighs the sparsity cost, so making $s_e$ larger helps minimize $-\mathcal{L}_{cls}$).

- **Decrease** $s_e$ if $Q_e > 0$, which means $\frac{\partial \mathcal{L}_{cls}}{\partial s_e} < \frac{\lambda}{m}$ (the loss increase from $s_e$ is less than the sparsity cost, or $s_e$ helps decrease loss, so making $s_e$ smaller helps minimize $J_{mask}$).

This aligns with the MaskNet's objective of minimizing $J_{mask}(\phi, \psi) = -\mathcal{L}_{cls}(\theta_{\text{fixed}}, \phi, \psi) + R(\phi, \psi)$. (Note: If using a hard-concrete gate or other mechanisms where the Jacobian $\partial\mathbf{s}/\partial\mathbf{p}$ might have more complex sign interactions, this direct interpretation would need adaptation.)

**Interaction Driving Robustness (Hypothesis):** The adversarial process unfolds as follows:

1. The MaskNet identifies edges $e$ where the current TaskNet $f_\theta$ is vulnerable (high positive $\frac{\partial \mathcal{L}_{cls}}{\partial s_e}$) or edges that are not sufficiently beneficial (low negative $\frac{\partial \mathcal{L}_{cls}}{\partial s_e}$) relative to the sparsity cost $\lambda/m$. It decreases the corresponding $s_e$ values.

2. The TaskNet receives this challenging mask $\mathbf{s}$ and updates its parameters $\theta$ via $\nabla_\theta \mathcal{L}_{cls}$ to minimize the loss despite the mask.

3. This adaptation by the TaskNet changes its internal representations and consequently alters the loss sensitivity $\nabla_{\mathbf{s}}\mathcal{L}_{cls}$ for the next MaskNet update.

We hypothesize that domain-specific, spurious edges exhibit high positive $\frac{\partial \mathcal{L}_{cls}}{\partial s_e}$ for a non-robust TaskNet. The MaskNet consistently targets these edges for down-weighting. The TaskNet, to survive, must adapt $\theta$ such that its predictions become less dependent on these specific edges, thereby reducing the magnitude of $\frac{\partial \mathcal{L}_{cls}}{\partial s_e}$ for those spurious edges. This forces the TaskNet to rely more heavily on other structural patterns, potentially the domain-invariant ones, whose presence consistently leads to a decrease in loss (negative $\frac{\partial \mathcal{L}_{cls}}{\partial s_e}$) even under perturbation. The gradient dynamics thus provide a mechanism pushing the TaskNet towards representations robust to the removal of likely domain-specific structural features identified by the MaskNet.

## F    Additional Ablation Studies: Hyperparameters for Graph Enrichment

This section details further ablation studies on hyperparameters related to the graph enrichment process (kNN and Spectral clustering edges) used in EdgeMask-DG*. All experiments, unless otherwise specified, use the default hyperparameters outlined in the main paper. Leave-one-out cross-validation across the three citation datasets ('acmv9', 'citationv1', 'dblpv7') is used for these hyperparameter sweeps. (Results are from single runs; std. dev. over multiple seeds would be beneficial for robustness assessment.)

### F.1    kNN K Ablation

We study the effect of varying the number of neighbors ('knnk') used for constructing the kNN feature graph within our full EdgeMask-DG*(Spec+kNN) method. The spectral K is fixed at 100. Average results over the three leave-one-out scenarios are presented in Table 11 and Figure 2.

Table 11: kNN K ablation results (average over 3 Scenarios). Performance metrics (Avg Micro-F1, Avg Macro-F1) are reported. The default value (k=10) and the best average performance (k=3) are highlighted.

| kNN K | Avg Micro-F1 | Avg Macro-F1 |
|---|---|---|
| 3.0000 | 0.7383 | 0.7260 |
| 5.0000 | 0.7263 | 0.7115 |
| 10.0000 | 0.7307 | 0.7175 |
| 20.0000 | 0.6871 | 0.6428 |
| 50.0000 | 0.6362 | 0.5166 |

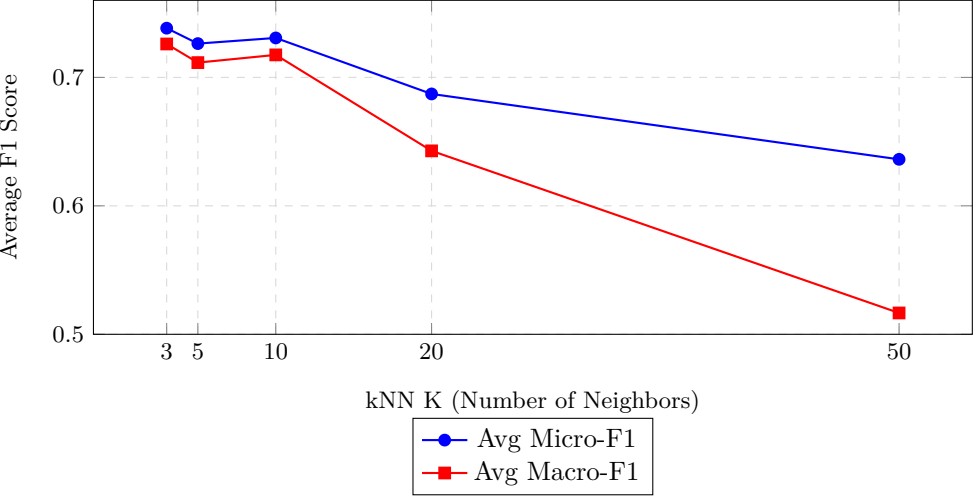

Figure 2: Average performance vs. kNN K (number of neighbors). Results averaged over 3 leave-one-out scenarios.

**Analysis:** The choice of 'knnk' significantly impacts performance. Figure 2 shows a clear trend: performance is best for small values of K (peaking at K=3 on average), remains relatively good at K=10 (our default), but drops sharply for larger values (K=20 and K=50). The particularly steep decline in Macro-F1 for K=50 suggests that including too many neighbors introduces excessive noise or spurious connections, potentially linking nodes across class boundaries and harming the classification of minority classes. This indicates that the kNN edges are most beneficial when they capture strong, local similarities based on node features. While K=3 yields the best average, K=10 provides a good balance and strong performance, justifying its use as a default, although smaller values might offer further improvements.

### F.2 Spectral K Ablation

We analyze the effect of varying the number of clusters ('spectralk') used for generating spectral feature edges in our full EdgeMask-DG*(Spec+kNN) method. The kNN K is fixed at 10. Average results over the three leave-one-out scenarios, including training time, are presented in Table 12 and Figure 3.

Table 12: Spectral K ablation results (average over 3 Scenarios, except k=50). Performance metrics (Avg Micro-F1, Avg Macro-F1) and average training time (s) are reported. The default (k=100) and best performing (k=200) values are highlighted.

| Spectral K | Avg Micro-F1 | Avg Macro-F1 | Avg Train Time (s) |
|---|---|---|---|
| 50.0000 | 0.7126 | 0.6551 | 393.30 |
| 100.0000 | 0.7463 | 0.7354 | 363.0800 |
| 150.0000 | 0.7404 | 0.7268 | 323.6400 |
| 200.0000 | 0.7498 | 0.7364 | 309.3200 |
| 250.0000 | 0.7457 | 0.7339 | 325.2400 |
| 300.0000 | 0.7449 | 0.7301 | 318.8800 |

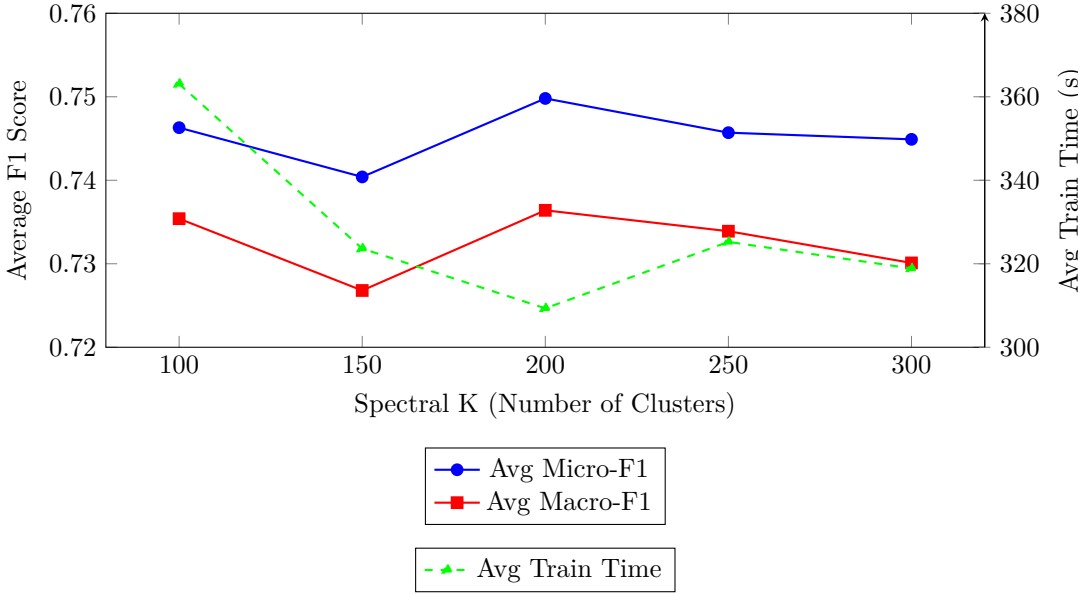

Figure 3: Average performance and training time vs. spectral $K$. F1 scores (left axis) and training time (right axis) averaged over three leave-one-out runs (note: $k = 50$ omitted due to OOM).

