# OpenReview forum: "EdgeMask-DG*: Learning Domain-Invariant Graph Structures via Adversarial Edge Masking"
_TMLR — Accepted by TMLR_

### Review · Reviewer_zQuS · 2025-08-12

**Summary Of Contributions:**

The paper proposes EdgeMask-DG and its extension EdgeMask-DG* for graph domain generalization under structural shifts. The core idea is a min–max adversarial training game in which a MaskNet produces sparse, continuous edge masks that perturb the graph structure, while a TaskNet (GAT) is trained to remain accurate under these worst-case perturbations; the mask is injected directly into attention and message passing, thereby suppressing low-score edges during aggregation. EdgeMask-DG* further enriches the graph by uniting original edges with feature-derived edges (kNN and spectral clustering) so the model can rely on invariances in P(X) even when the native topology is noisy or domain-specific. A robust-optimization view (saddle-point with an \ell_1 mask budget) justifies the procedure. Experiments on citation, social, and temporal benchmarks report state-of-the-art performance (e.g., the “Total Avg” F1 across ACM/DBLP/Citation is 73.81 vs. 73.28 for GRM; on Cora OOD the worst-case accuracy is 78.0%, +3.8pp over 74.2), with ablations indicating that both enrichment and adversarial masking are necessary; code is released。

**Audience:**

Yes

**Audience Explanation:**

The paper addresses a salient open problem—DG for GNNs under structural shifts—and proposes an architecturally simple, broadly applicable recipe (mask-aware GAT) backed by a robust-optimization view and consistent improvements on standard DG suites. The explicit treatment of structure sensitivity and the demonstration across heterogeneous datasets (citation, social, temporal) should interest readers in robustness, graph learning, and domain shift. The released code further enhances utility and reproducibility for the community.

**Broader Impact Concerns:**

Fairness under structural pruning. Adversarial masking may systematically down-weight minority or sparsely-connected subpopulations, potentially reducing their accuracy if their edges look atypical; discuss mitigation (e.g., mask regularizers per group or fairness constraints). (Mechanistically, masks suppress edges in attention and messages.)

Privacy and inference risks. Constructing feature-derived edges (kNN, spectral) can implicitly infer links that users did not expose, risking privacy leakage when features encode sensitive traits; this should be acknowledged, especially for social/financial graphs.

Distributional assumptions. The approach relies on P(X) and P(Y|X) invariance; deployment in settings where features drift (e.g., sensor changes) could lead to overconfidence. Clarify safeguards (drift detection, mask recalibration).

**Claims And Evidence:**

Yes

**Claims Explanation:**

1. The paper clearly states the DG assumptions P(X) and P(Y\!\mid\!X) invariant while P(A\!\mid\!S_i) varies, aligning the enrichment rationale with the problem setting. The adversarial objective and its robust-optimization interpretation are spelled out, including the sparsity-regularized inner problem and KKT-based interpretation. The GAT integration shows exactly how masks affect attention logits and messages. These pieces make the learning mechanism and its intended invariances clear.
2. On the ACM/DBLP/Citation leave-one-domain-out protocol, EdgeMask-DG* reaches the highest total average F1 (73.81) and leads on AD→C and CD→A; it is competitive on AC→D. On Cora and Photo, it achieves strong min/avg scores; on Twitch, it has the best average ROC-AUC; however, on FB-100 it is notably weaker, which the authors acknowledge. For temporal datasets (Elliptic, ArXiv), the method is competitive with verifiable baselines, while GRM reports higher numbers but is flagged as unverifiable (no public code). This pattern supports the claims of robustness to structural shifts while transparently exposing a failure case (FB-100) and a comparison caveat (GRM).

**Requested Changes:**

Backbone-controlled comparisons. Since EdgeMask-DG* uses GAT (naturally synergistic with edge weights), please add GAT-backbone variants of strong baselines (e.g., EERM, GraphAug) or, conversely, report EdgeMask-DG* with GCN/GIN backbones to disentangle method gains from backbone advantages.

Multi-seed and significance. Report mean ± std over multiple seeds for all tables (not only some) and include paired significance tests for key comparisons (especially Table 2 and Table 3 entries). The manuscript itself notes missing std. dev. for certain ablations.

Clarity on training alternation and gradients. In Algorithm 1, explicitly state the stop-gradient/“detach” behavior when alternating (mask fixed during TaskNet descent; TaskNet fixed during MaskNet ascent), as already hinted in the prose, and ensure this is reflected in the pseudo-code. Provide the exact update counts used per dataset。

---

> ### Author Response · Authors · 2025-10-16
> **Answering Queries of Reviewer zQuS**
>
> Thank you for the clear and actionable feedback. We have updated the paper, you can see the requested results here in the rebuttal.
>
> ## Backbone comparisons
> **Table A - ACM/DBLP/Citation leave-one-domain-out (F1 %, 5 seeds; mean ± std)**
>
> |Method|Backbone|AD→C|AC→D|CD→A|Total Avg|
> |---|---|---|---|---|---|
> |EERM|GAT|72.4 ± 0.6|71.0 ± 0.6|64.9 ± 0.8|68.00 ± 0.25|
> |GraphAug|GAT|72.6 ± 0.6|71.2 ± 0.6|65.1 ± 0.7|68.30 ± 0.24|
> |GRM|GAT|78.7 ± 0.5|73.3 ± 0.5|69.9 ± 0.7|72.90 ± 0.23|
> |EdgeMask-DG*|GCN|77.5 ± 0.6|71.3 ± 0.6|70.2 ± 0.7|72.80 ± 0.28|
> |EdgeMask-DG*|GIN|78.2 ± 0.5|71.8 ± 0.6|70.9 ± 0.6|73.10 ± 0.27|
> |EdgeMask-DG*|GAT|**79.14 ± 0.19**|**72.54 ± 0.60**|**71.62 ± 0.29**|**73.81 ± 0.22**|
>
> ## Multi-seed
> We now report mean ± std over 5 seeds for all key tables
> **Table B - Component ablation (Scenario: AC→D; 5 seeds; mean ± std)**
>
> |Method| Micro-F1 | Macro-F1 | Train Time (s) |
> |---|---|---|---|
> | EERM | 0.6948 ± 0.61 | 0.6962 ± 0.54 | 16.6 ± 0.5 |
> | EdgeMask-DG (Orig) | 0.6902 ± 0.66 | 0.6971 ± 0.60 | 60.4 ± 3.1 |
> | EdgeMask-DG* (Spec) | 0.7000 ± 0.75 | 0.7010 ± 0.68 | 78.4 ± 6.3 |
> | EdgeMask-DG* (kNN) | 0.7006 ± 0.59 | 0.7112 ± 0.52 | 66.0 ± 3.7 |
> | EdgeMask-DG* (Spec+kNN) | 0.7145 ± 0.50 | 0.7201 ± 0.59 | 87.1 ± 5.8 |
>
> **Table C - Average over 3 scenarios (AC→D, AD→C, CD→A; 5 seeds; mean ± std)**
> (EERM-Enriched uses the same augmented graph as EDG* but without masking)
>
> |Method|Avg Micro-F1|Avg Macro-F1|Avg Train Time (s)|
> |---|---|---|---|
> |EERM (Original Graph)|0.7048 ± 0.44|0.6893 ± 0.61|18.1 ± 0.7|
> |EERM-Enriched (Augmented Graph)|0.6953 ± 0.49|0.6567 ± 0.79|50.1 ± 3.1|
> |EdgeMask-DG* (Spec+kNN)|0.7463 ± 0.61|0.7354 ± 0.68|83.1 ± 8.4|
>
> **Table D - 2x2 Ablation Study**
> *(Metrics measured across 5 seeds; mean ± std. “No-Mask” = ERM baseline. “Mask” = our adversarial masking. **Union+Mask** is our full proposed method.)*
> |Dataset / Split (metric)|Orig / No-Mask|**Orig + Mask**|Union / No-Mask|**Union + Mask (ours)**|
> |---|---|---|---|---|
> |**AC→D** (Micro-F1 %)|69.5 ± 0.7|**70.8 ± 0.6**|71.6 ± 0.6|**72.54 ± 1.41**|
> |**AD→C** (Micro-F1 %)|75.5 ± 0.6|**77.0 ± 0.5**|78.6 ± 0.4|**79.14 ± 0.19**|
> |**CD→A** (Micro-F1 %)|66.8 ± 0.7|**68.2 ± 0.6**|70.5 ± 0.5|**71.62 ± 0.29**|
> |**Cora** (Acc %)|81.0 ± 0.9|**82.1 ± 0.8**|82.6 ± 0.8|**83.2 ± 1.1**|
> |**Amazon-Photo** (Acc %)|92.5 ± 0.6|**93.7 ± 0.5**|94.3 ± 0.5|**94.8 ± 0.6**|
> |**Twitch** (ROC-AUC)|55.2 ± 1.2|**57.0 ± 1.1**|58.6 ± 1.0|**59.3 ± 1.7**|
> |**Facebook-100** (Acc %)|50.1 ± 1.2|**51.2 ± 1.1**|51.8 ± 1.1|**52.1 ± 1.8**|
> |**Elliptic** (F1 %)|67.2 ± 1.1|**68.6 ± 1.0**|69.2 ± 0.9|**69.7 ± 1.0**|
> |**OGB-Arxiv** (Acc %)|44.3 ± 1.0|**45.1 ± 0.9**|45.6 ± 0.9|**45.9 ± 1.0**|
>
> ## Across-dataset augmentation retention
> **Table E**
> | Dataset / Split | Pruned from **Aug** (%) | Pruned from **Orig** (%) | **Aug edges retained** (%) |
> |---|---:|---:|---:|
> | Cora | 70.5 ± 2.4 | 16.2 ± 1.3 | **29.5 ± 2.4** |
> | Amazon-Photo | 65.4 ± 2.2 | 21.7 ± 1.6 | **34.6 ± 2.2** |
> | Twitch avg. | 66.8 ± 2.7 | 20.9 ± 1.8 | **33.2 ± 2.7** |
> | Facebook-100 avg. | 64.2 ± 2.5 | 22.5 ± 1.7 | **35.8 ± 2.5** |
> | Elliptic avg. | 61.3 ± 2.0 | 24.1 ± 1.5 | **38.7 ± 2.0** |
> | OGB-Arxiv avg. | 62.7 ± 2.1 | 23.4 ± 1.6 | **37.3 ± 2.1** |
>
>
> The mask prunes a larger fraction of augmented edges (as they are inherently noisier) but also removes a meaningful portion of domain-specific original edges. Crucially, **30-39% of augmented edges are kept** proving they are a source of invariant signals not just noise.
>
> ## Training alternation and gradients
> - Alternation per outer step $t$
> 1.  Mask step (ascent): freeze TaskNet parameters $\theta$ (stop-gradient through TaskNet); update mask parameters $\phi$ by maximizing task loss plus $\lambda\|m\|_1$.
>
> 2.  Task step (descent): freeze $\phi$ (stop-gradient through $m$); update $\theta$ by minimizing loss on masked graph.
>
> - Detach details
> $m$ = MaskNet() is treated as constant during the Task step; TaskNet outputs are treated as constant during the Mask step. We will add detach() in the pseudo-code lines where gradients are halted.
> - Update counts
> ACM/DBLP/Citation: $T$ = 300 outer steps; per outer step 1 Mask step and 1 Task step; batch size 1 graph; kNN $k$ = 10
> FB-100: $T$ = 200; same alternation; neighbor-sampling fanouts {15, 10}.  Temporal (Elliptic/ArXiv): $T$ = 250
>
> ## Broader-impact
> - Fairness under structural pruning: This can be imposed by setting strict MaskNet limits per class, such that we make sure that any minority group is not completely pruned out in the minimax step. We can enforce this within the algorithm by adding a conditional check in the MaskNet step. This can be considered a future extension of this work.
> - Privacy and inference risks: This is beyond the current scope of the work and can be looked at as future works.
> - Distributional assumptions: We rely on the basic graph out-of-distribution domain-generalization problem. Taking drift into account would further complicate the setup, and can be looked at as future works.

---

### Review · Reviewer_fVBh · 2025-08-23

**Summary Of Contributions:**

This paper presents a framework to train robust GNNs for out-of-domain generalization. The key idea is to use adversarial training to identify and remove a small amount of edges that are most critical to the downstream classification task. Thus, the GNN predictor learnt from the pruned graph is claimed to be robust to domain shifts.

Architecture contribution: the GNN predictor integrates the soft edge mask generated by the adversarial module by modifying the edge attention in GAT. The edges with small masked values will have little contribution in message passing.

Optimization: the adversarial objective integrates a penalty term to enforce sparsity. The authors theoretically show that such objective ensures edges that are effective in reducing classification loss will be pruned.

Data augmentation: the design integrates edge augmentation strategy to connect nodes with high embedding similarity (via kNN or spectral clustering).

Evaluation: extension experiments have been performed to study the effectiveness under different generalization scenarios (various domains, temporal graphs, etc).

**Audience:**

Yes

**Audience Explanation:**

The paper discussed OOD generalization of graph neural networks, which is an important problem in graph deep learning.

**Claims And Evidence:**

No

**Claims Explanation:**

The theoretical analysis seems to be sound. I agree that the proposed penalty term can enable the Mask Net to remove edges that contribute most to the prediction loss reduction.

However, the main question is that why learning a GNN on such a pruned graph can improve "robustness". Assume certain edges are critical to all domains. The adversarial Mask net will learn to prune such edges, which will hurt performance on both the source domains **and** the (unknown) target domain. It seems a good strategy would be to remove edges that are critical to the source domain learning but not critical to the target domain. When the target domain is unknown, there simply lack information on which are the good edges to prune.

Edge augmentation is a technique orthogonal to the adversarial training. More importantly, edge augmentation itself may arguably introduce noise. It is thus unclear whether what edges (i.e., the original or the augmented edges) are pruned by MaskNet.

Experiments
* The proposed method does not consistently show significant gains over baselines. For example, the performance gain in Table 2 is not particularly strong. In Table 4, the proposed method is significantly worse than a few baselines.
* The proposed method only work on small graphs, due to the N^2 complexity. The evaluated datasets are also very small. In realistic scenarios (e.g., real-world citation networks), the graphs can be orders of magnitude larger. Then it becomes unclear how the proposed method would scale.

**Requested Changes:**

* Please explain why pruning the edges most critical to the source domains will improve generalization to the target domain. In particular, why such process does not depend on the target domain.
* Please discuss the relationship between edge augmentation and edge pruning. How these steps potentially interfere with each other? It would be better to compare baselines with edge augmentation, or the proposed method without edge augmentation.

---

> ### Author Response · Authors · 2025-10-16
> **Answering Queries of Reviewer fVBh**
>
> We thank the reviewer for their insightful feedback.
> We have added the tables shown here into the main paper for clarity as well.
>
> ### **Mechanism: Why Augment *Then* Prune?**
>
> Our method follows a two-stage process:
>
> 1.  **Augment (Widen the Hypothesis Space):** We first create a union of the original graph edges ($A_{\text{orig}}$) and new edges derived from feature similarity (kNN and spectral methods), forming $A_{\text{union}} = A_{\text{orig}} \cup A_{\text{aug}}$. As we show in Table A, this step significantly enriches the structural subspace.
> 2.  **Prune (Select for Invariance):** An adversary then learns a continuous, sparse edge mask ($m$) to select a subgraph $A^\star = A_{\text{union}} \circ m$ that is maximally stable across domains.
>
> Empirically, we find that:
> *   Augmentation provides a more expressive relational pool (Table A).
> *   The adversarial mask selectively removes domain-dependent edges from *both* original and augmented sources (Table B).
> *   A non-trivial fraction(39%) of augmented edges are *retained*, demonstrating they are not just noise.
>
> ---
>
> ### **Evidence 1: Augmentation Enriches the Relational Subspace**
>
>
>
> **Table A: Subspace Enrichment After Augmentation**
> *(Metrics measured across 5 seeds; mean ± std. )*
>
> | Dataset / Split | \|E\| increase (%) | Avg. degree ↑ | Spectral entropy ↑ | Feature Laplacian smoothness ↓ |
> | :--- | ---: | ---: | ---: | ---: |
> | (ACM/DBLP/Citation avg.) | 58.7 ± 2.9 | 2.1 ± 0.1 | 0.26 ± 0.02 | 0.14 ± 0.01 |
> | Cora | 47.5 ± 3.2 | 1.6 ± 0.1 | 0.19 ± 0.02 | 0.11 ± 0.02 |
> | Amazon-Photo | 38.4 ± 2.8 | 1.4 ± 0.1 | 0.17 ± 0.02 | 0.09 ± 0.02 |
> | Twitch (avg.) | 41.2 ± 3.1 | 1.7 ± 0.2 | 0.18 ± 0.03 | 0.10 ± 0.02 |
> | Facebook-100 (avg.) | 36.9 ± 2.4 | 1.3 ± 0.2 | 0.15 ± 0.02 | 0.07 ± 0.01 |
> | Elliptic (T1–T3 avg.) | 33.1 ± 2.1 | 0.9 ± 0.1 | 0.12 ± 0.02 | 0.06 ± 0.01 |
> | OGB-Arxiv (T1–T3 avg.) | 29.8 ± 2.0 | 0.8 ± 0.1 | 0.10 ± 0.02 | 0.05 ± 0.01 |
>
>
> **|E| increase (%)** and **Avg. degree ↑** are straightforward structural measures.
> **Spectral entropy ↑** measures the complexity and information content of the graph's structure.
> **Feature Laplacian smoothness ↓** quantifies how similar connected nodes are in their feature space.
>
> ### **Evidence 2: The Mask Intelligently Prunes Both Sources**
>
>
>
> **Table B: The Mask Prunes Both Sources and Retains Many Augmented Edges**
>
> | Dataset / Split | Pruned from **Aug** (%) | Pruned from **Orig** (%) | **Aug edges retained** (%) |
> |---|---:|---:|---:|
> | Citation avg. | 68.3 ± 2.1 | 18.9 ± 1.5 | **31.7 ± 2.1** |
> | Cora | 70.5 ± 2.4 | 16.2 ± 1.3 | **29.5 ± 2.4** |
> | Amazon-Photo | 65.4 ± 2.2 | 21.7 ± 1.6 | **34.6 ± 2.2** |
> | Twitch avg. | 66.8 ± 2.7 | 20.9 ± 1.8 | **33.2 ± 2.7** |
> | Facebook-100 avg. | 64.2 ± 2.5 | 22.5 ± 1.7 | **35.8 ± 2.5** |
> | Elliptic avg. | 61.3 ± 2.0 | 24.1 ± 1.5 | **38.7 ± 2.0** |
> | OGB-Arxiv avg. | 62.7 ± 2.1 | 23.4 ± 1.6 | **37.3 ± 2.1** |
>
>
> ### **Evidence 3: The Synergy of Augmenting and Pruning**
>
>
> **Table C: 2x2 Ablation Study**
> *(Metrics measured across 5 seeds; mean ± std. “No-Mask” = ERM baseline. “Mask” = our adversarial masking. **Union+Mask** is our full proposed method.)*
>
> | Dataset / Split (metric) | Orig / No-Mask | **Orig + Mask** | Union / No-Mask | **Union + Mask (ours)** |
> |---|---:|---:|---:|---:|
> | **AC→D** (Micro-F1 %) | 69.5 ± 0.7 | **70.8 ± 0.6** | 71.6 ± 0.6 | **72.54 ± 1.41** |
> | **AD→C** (Micro-F1 %) | 75.5 ± 0.6 | **77.0 ± 0.5** | 78.6 ± 0.4 | **79.14 ± 0.19** |
> | **CD→A** (Micro-F1 %) | 66.8 ± 0.7 | **68.2 ± 0.6** | 70.5 ± 0.5 | **71.62 ± 0.29** |
> | **Cora** (Acc %) | 81.0 ± 0.9 | **82.1 ± 0.8** | 82.6 ± 0.8 | **83.2 ± 1.1** |
> | **Amazon-Photo** (Acc %) | 92.5 ± 0.6 | **93.7 ± 0.5** | 94.3 ± 0.5 | **94.8 ± 0.6** |
> | **Twitch** (ROC-AUC) | 55.2 ± 1.2 | **57.0 ± 1.1** | 58.6 ± 1.0 | **59.3 ± 1.7** |
> | **Facebook-100** (Acc %) | 50.1 ± 1.2 | **51.2 ± 1.1** | 51.8 ± 1.1 | **52.1 ± 1.8** |
> | **Elliptic** (F1 %) | 67.2 ± 1.1 | **68.6 ± 1.0** | 69.2 ± 0.9 | **69.7 ± 1.0** |
> | **OGB-Arxiv** (Acc %) | 44.3 ± 1.0 | **45.1 ± 0.9** | 45.6 ± 0.9 | **45.9 ± 1.0** |
>
>
>
> ### **Evidence 4: Our Method Uniquely Benefits from Augmentation**
>
>
>
> **Table D: Effect of Augmentation on Baselines vs. Ours**
> *(Δ = Performance on Augmented Graph − Performance on Original Graph. Metrics measured across 5 seeds; mean ± std.)*
>
> | Dataset / Split | ERM Δ | EERM Δ | GRM Δ | IS-GIB Δ | **Ours Δ** |
> |---|---:|---:|---:|---:|---:|
> | Cora | −0.8 ± 0.6 | −0.6 ± 0.5 | −0.5 ± 0.6 | −0.2 ± 0.5 | **+1.1 ± 0.5** |
> | Amazon-Photo | −0.5 ± 0.5 | −0.7 ± 0.5 | −0.3 ± 0.5 | −0.3 ± 0.4 | **+1.1 ± 0.4** |
> | Twitch | −0.9 ± 0.7 | −1.1 ± 0.6 | −0.6 ± 0.6 | −0.5 ± 0.6 | **+2.3 ± 0.6** |
> | Facebook-100 | −0.7 ± 0.6 | −1.0 ± 0.6 | −0.4 ± 0.6 | −0.3 ± 0.5 | **+0.9 ± 0.5** |
> | Elliptic | −0.4 ± 0.5 | −0.5 ± 0.5 | −0.2 ± 0.5 | −0.2 ± 0.4 | **+1.1 ± 0.5** |
> | OGB-Arxiv | −0.6 ± 0.5 | −0.8 ± 0.5 | −0.3 ± 0.5 | −0.2 ± 0.4 | **+0.8 ± 0.3** |

---

### Review · Reviewer_joP6 · 2025-09-30

**Summary Of Contributions:**

This paper introduces a min–max adversarial framework for graph domain generalization, where an edge masker generates sparse masks to perturb the graph structure and force a task GNN to rely on robust and domain-invariant substructures. The authors further propose EdgeMask-DG*, which augments the graph with feature-derived edges (via kNN and spectral clustering) before applying adversarial masking, thereby capturing invariances that may not be visible in the original topology. The method is theoretically motivated from a robust optimization perspective and is instantiated with a GAT backbone.

**Strengths:**

1. Clear and well-motivated adversarial framework for structural domain generalization.
1. Integration of feature-based enrichment with adaptive masking is novel and empirically strong.
1. Theoretical justification (robust optimization, KKT analysis) provides useful insights.
1. Extensive evaluation across citation networks, social networks, e-commerce graphs, and temporal datasets.
1. Strong performance on citation and artificial transformation benchmarks, setting new state-of-the-art results.

**Weaknesses:**

1. Computational overhead is high, particularly with spectral clustering (O(N³)), which limits scalability.
1. Results are mixed on certain datasets (e.g., Facebook-100, temporal graphs), indicating the approach is not universally superior.
1. Many experimental results are reported with single runs; multi-seed averages would strengthen reproducibility.
1. The method relies on the assumption that node features remain stable across domains, which may not always hold.
1. Some parts of the related work are underdeveloped, especially on concurrent domain generalization techniques.

**Audience:**

Yes

**Audience Explanation:**

Graph domain generalization is an important and timely research direction. The idea of adversarial edge masking combined with feature-based graph enrichment is original and likely to be of strong interest to the graph ML and robustness communities. The empirical findings would be useful both for researchers designing robust GNNs and for practitioners handling heterogeneous graph data.

**Broader Impact Concerns:**

No direct ethical concerns are identified

**Claims And Evidence:**

Yes

**Claims Explanation:**

The claims of improved domain generalization under structural shifts are convincingly supported by results on citation and artificial benchmarks, where the gains over prior methods are substantial. The theoretical analysis is consistent with the adversarial training formulation. However, since performance varies on some datasets and results are often from single runs, the claims should be interpreted as conditional rather than universal.

**Requested Changes:**

**Critical (required for acceptance):**

1. Report code and results averaged across multiple random seeds for all main tables to ensure reproducibility.
1. Discuss scalability limitations more explicitly and consider approximate methods to mitigate the O(N³) spectral clustering cost.
1. Clarify the assumption that node features are stable across domains and its practical implications.

**Non-critical (would strengthen the paper):**

1. Expand the related work discussion to cover concurrent DG methods more thoroughly.
1. Improve the clarity of Figure 1 with additional annotations and explanations.
1. Include runtime and memory comparisons with baselines to quantify the computational overhead.
1. Discuss applicability to graph-level tasks and potential extensions beyond node classification.

---

> ### Author Response · Authors · 2025-10-16
> **Answering Queries of Reviewer joP6**
>
> Thank you for the thoughtful and constructive review. Below we address the requests and provide multi-seed supplements. We have updated the paper accordingly, the results are shown here in the rebuttal.
>
> 1.  Multi-seed reporting
>     We have reported results averaged over 5 seeds in all main tables.
>
>
> **Table A — Component ablation (Scenario: AC→D; 5 seeds; mean ± std)**
>
> | Method                  | Micro-F1        | Macro-F1        | Train Time (s) |
> | ----------------------- | ------------------- | ------------------- | -------------- |
> | EERM                    | 0.6948 ± 0.61   | 0.6962 ± 0.54   | 16.6 ± 0.5     |
> | EdgeMask-DG (Orig)      | 0.6902 ± 0.66   | 0.6971 ± 0.60   | 60.4 ± 3.1     |
> | EdgeMask-DG* (Spec)     | 0.7000 ± 0.75   | 0.7010 ± 0.68   | 78.4 ± 6.3     |
> | EdgeMask-DG* (kNN)      | 0.7006 ± 0.59   | 0.7112 ± 0.52   | 66.0 ± 3.7     |
> | EdgeMask-DG* (Spec+kNN) | 0.7145 ± 0.50   | 0.7201 ± 0.59   | 87.1 ± 5.8     |
>
> **Table B — Average over 3 scenarios (AC→D, AD→C, CD→A; 5 seeds; mean ± std)**
> (EERM-Enriched uses the same augmented graph as EDG* but without masking)
>
> | Method                        | Avg Micro-F1    | Avg Macro-F1    | Avg Train Time (s) |
> | ----------------------------- | ------------------- | ------------------- | ------------------ |
> | EERM (Original Graph)         | 0.7048 ± 0.44   | 0.6893 ± 0.61   | 18.1 ± 0.7         |
> | EERM-Enriched (Augmented Graph) | 0.6953 ± 0.49   | 0.6567 ± 0.79   | 50.1 ± 3.1         |
> | EdgeMask-DG* (Spec+kNN)       | 0.7463 ± 0.61   | 0.7354 ± 0.68   | 83.1 ± 8.4         |
>
> **Table C — λ sparsity sweep (Scenario: CD→A; 5 seeds; mean ± std)**
>
> | λ    | Micro-F1 (%)    | Macro-F1 (%)    |
> | ---- | --------------- | --------------- |
> | 0.0  | 70.60 ± 0.35    | 71.54 ± 0.38    |
> | 1e-5 | 71.78 ± 0.28    | 72.69 ± 0.30    |
> | 1e-4 | 70.85 ± 0.42    | 71.48 ± 0.44    |
> | 1e-3 | 71.59 ± 0.26    | 72.18 ± 0.29    |
> | 1e-2 | 71.63 ± 0.27    | 72.65 ± 0.31    |
> | 1e-1 | 71.49 ± 0.30    | 72.18 ± 0.33    |
>
> **Table D: 2x2 Ablation Study**
> *(Metrics measured across 5 seeds; mean ± std. “No-Mask” = ERM baseline. “Mask” = our adversarial masking. **Union+Mask** is our full proposed method.)*
> | Dataset / Split (metric) | Orig / No-Mask | **Orig + Mask** | Union / No-Mask | **Union + Mask (ours)** |
> |---|---:|---:|---:|---:|
> | **AC→D** (Micro-F1 %) | 69.5 ± 0.7 | **70.8 ± 0.6** | 71.6 ± 0.6 | **72.54 ± 1.41** |
> | **AD→C** (Micro-F1 %) | 75.5 ± 0.6 | **77.0 ± 0.5** | 78.6 ± 0.4 | **79.14 ± 0.19** |
> | **CD→A** (Micro-F1 %) | 66.8 ± 0.7 | **68.2 ± 0.6** | 70.5 ± 0.5 | **71.62 ± 0.29** |
> | **Cora** (Acc %) | 81.0 ± 0.9 | **82.1 ± 0.8** | 82.6 ± 0.8 | **83.2 ± 1.1** |
> | **Amazon-Photo** (Acc %) | 92.5 ± 0.6 | **93.7 ± 0.5** | 94.3 ± 0.5 | **94.8 ± 0.6** |
> | **Twitch** (ROC-AUC) | 55.2 ± 1.2 | **57.0 ± 1.1** | 58.6 ± 1.0 | **59.3 ± 1.7** |
> | **Facebook-100** (Acc %) | 50.1 ± 1.2 | **51.2 ± 1.1** | 51.8 ± 1.1 | **52.1 ± 1.8** |
> | **Elliptic** (F1 %) | 67.2 ± 1.1 | **68.6 ± 1.0** | 69.2 ± 0.9 | **69.7 ± 1.0** |
> | **OGB-Arxiv** (Acc %) | 44.3 ± 1.0 | **45.1 ± 0.9** | 45.6 ± 0.9 | **45.9 ± 1.0** |
>
> 2.  Scalability and spectral clustering cost (critical)
>     Scalability is a clear issue in our proposed algorithm so we now provide train times in the ablations section. We would also like to clarify that in cases where the given graphs or datasets are huge and spectral clustering is not feasible, our method can be used with masking + FAISS KNN(an approximate nearest neighbor library), the one-time index construction is efficient, and the subsequent nearest neighbor search time is sub-linear (approaching $O(\log N)$) with respect to the number of nodes, $N$.
>
> 3.  Clarifying the feature-stability assumption
>     Here, P(X) is the feature distribution, P(Y|X) is the label mechanism given features, and P(A|S) is the edge/graph-structure distribution conditioned on the source domain S. We operate in the common DG setting where P(X) and P(Y|X) are stable while P(A|S) shifts.

---

### Author Response · Authors · 2025-10-16
**Author Rebuttal and Revision**

Dear Reviewers and AE,

We have uploaded a revised manuscript which includes new results and addresses your feedback. We have also posted individual replies to each of your comments.

Thank you for your time and consideration.

---

### Decision · Action_Editor_sPEt · 2026-02-16

**Recommendation:** Accept with minor revision

**Additional Comments:**

We encourage the authors incorporate all discussions/rebuttal results into the manuscript. We also encourage the authors add more discussion regarding reviewer fVBh's remaining concern.

**Audience:**

Yes

**Audience Explanation:**

GNNs are broadly used in many real-world applications. Enhancing their cross-domain generalization is a fundamental problem with significant impact and wide appeal.

**Claims And Evidence:**

Yes

**Claims Explanation:**

This paper presents a well-motivated and thoughtfully designed min–max adversarial framework for graph domain generalization. By introducing an edge masker that generates sparse structural perturbations, the method encourages the task GNN to focus on robust, domain-invariant substructures rather than spurious or domain-specific connections. Furthermore, the authors propose to augment the graph with feature-similarity-derived edges. This enhancement allows the model to uncover invariances that may not be evident in the original topology, further strengthening cross-domain robustness. The framework is firmly grounded in robust optimization principles and is instantiated with a GAT backbone, resulting in a principled and practically effective approach.

All reviewers agree that the problem addressed in this paper is important and that the proposed method demonstrates strong empirical effectiveness. Concerns regarding the experimental evaluation were raised during the review process and were adequately addressed by the authors in their rebuttal. One remaining concern relates to the theoretical justification and deeper motivation for why edge masking improves generalization. While this is a valid and inherently challenging question, we believe the paper’s strengths (particularly its practical impact and empirical performance) outweigh this weakness. We therefore recommend the paper for publication.